# Development and evolution of an anomalous Asian dust event across Europe in March 2020

Laura Tositti[1], Erika Brattich[2], Claudio Cassardo[3], Pietro Morozzi[1], Alessandro Bracci[2], Angela Marinoni[4], Silvana Di Sabatino[2], Federico Porcù[2], Alessandro Zappi[1]

[1] Department of Chemistry "Giacomo Ciamician", Alma Mater Studiorum University of Bologna, Via F. Selmi, 2, 40126 Bologna (Italy)

[2] Department of Physics and Astronomy "Augusto Righi", Alma Mater Studiorum University of Bologna Viale Berti Pichat 6/2, 40127 Bologna (Italy)

[3] Department of Physics "Amedeo Avogadro", via P. Giuria 1, 10125 Torino (Italy)

[4] National Research Council of Italy, Institute of Atmospheric Sciences and Climate (CNR-ISAC), Via P. Gobetti, 101, 40129 Bologna (Italy)

*Correspondence to*: Laura Tositti (laura.tositti@unibo.it)

**Abstract.** This paper concerns an in-depth analysis of an exceptional incursion of mineral dust over Southern Europe in late March 2020 (27-30 March 2020). This event was associated with an anomalous circulation pattern leading to several days of $PM_{10}$ exceedances in connection with a dust source located in Central Asia, a rare source of dust for Europe, more frequently affected by dust outbreaks from the Sahara desert. The synoptic meteorological configuration was analyzed in detail, while aerosol evolution during the transit of the dust plume over Northern Italy was assessed at high time resolution by means of optical particle counting at three stations, namely Bologna, Trieste, and Mt. Cimone, allowing to reveal transport timing among the three locations. Back-trajectory analyses supported by CAMS (Copernicus Atmosphere Monitoring Service) maps allowed to locate the mineral dust source area in the Aralkum region. The event was therefore analyzed by observing particle number size distribution with the support of chemical composition analysis. It is shown that $PM_{10}$ exceedance recorded is associated with a large fraction of coarse particles in agreement with mineral dust properties. Both in-situ number size distribution and vertical distribution of the dust plume were cross-checked by Lidar Ceilometer and AOD data from two nearby stations, showing that the dust plume, differently from those originated in the Sahara desert, traveled close to the ground up to a height of about 2 km. The limited mixing layer height caused by high concentrations of absorbing and scattering aerosols caused the mixing of mineral dust with other locally-produced ambient aerosols, thereby potentially increasing its morbidity effects.

## 1 Introduction

Mineral dust originating in desert regions is one of the main components of the ambient aerosol affecting air quality and human health, cloud formation, ocean ecosystems, and climate (Knippertz and Stuut, 2014; Schepanski, 2018). Despite its significance, mineral dust still represents one of the largest uncertainties in climate modeling (Adebiyi and Kok, 2020; IPCC Working Group I, 2013). Mineral dust is often transported at a large scale, influencing vast continental and ocean areas (Barkan and Alpert, 2010). The most immediate effect of mineral dust in a specific area is the increase, sometimes dramatic, of particulate matter (PM) mass loading (Brattich et al., 2015a), mainly in the coarse fraction. Several studies also demonstrated that mineral dust can affect cloud processing (Bangert et al., 2012) and biogeochemical cycles of ecosystems (Okin et al., 2004). Numerous studies documented the health effects of mineral dust transport in various parts of the world (Domínguez-Rodríguez et al., 2021; Fubini and Fenoglio, 2007; Keil et al., 2016; Sajani et al., 2011; Stafoggia et al., 2016), owing to the conjunct impact of mineralogy, allergens, and pathogens (García-Pando et al., 2014). For these reasons, it is essential to study mineral dust composition in connection with its atmospheric path, source region, physicochemical properties, and modifications associated with environmental, climatic, and health issues.

The most important dust-source regions at the global scale are the North African Sahara region, the Arabian desert, the central Asia deserts (the Kyzylkum in Uzbekistan and Kazakhstan, and the Karakum in Turkmenistan), the Gobi and the Taklamakan deserts in eastern Asia, the Dasht-e Margo (the desert region between Pakistan, Iran, and Afghanistan), the Great Basin (USA), the Kalahari desert (Southern Africa), and the central Australia deserts (Calidonna et al., 2020; Prospero et al., 2002; Washington et al., 2003). Recently another prominent source of mineral dust is drawing research interest, i.e., the semi-arid region between the Persian Gulf and the Caspian Sea. Several studies (Gholamzade Ledari et al., 2020; Kaskaoutis et al., 2018; Rashki et al., 2018) reported the transport of mineral dust originating in this region, usually towards central and eastern Asia. With the purpose of evaluating the wind regime and dust activity in this region, a particular index, the Caspian Sea – Hindu Kush Index (CasHKI), has been introduced and studied in a long-term time series in previous studies (Kaskaoutis et al., 2018). Previous works have suggested that the Caspian Sea high largely impacts dust activity and transport of dust plumes over central and south-west Asia (Hamidianpour et al., 2021; Kaskaoutis et al., 2016; Labban et al., 2021). In turn, the combination of the Caspian Sea high with the low over the North Sea creates the North Sea-Caspian Pattern (NCP) that significantly modulates climatic conditions and potentially even dust activity over the Middle East and the Mediterranean (Brunetti and Kutiel, 2011; Kutiel and Benaroch, 2002).

The primary source of desert dust reaching Southern Europe, including Italy, is the Sahara desert (Brattich et al., 2015a; Calidonna et al., 2020). Typically, Saharan Dust transports to the central Mediterranean occur during cyclonic conditions in winter, spring, and fall, and during anticyclonic conditions in summer (Gobbi et al., 2019). In general, dust storms to the central Mediterranean are more frequent in the spring-summer

periods, while these are rare in the cold seasons (Brattich et al., 2015a; Duchi et al., 2016) to late winter (Fubini and Fenoglio, 2007; IPCC Working Group I, 2013). Saharan dust outbreaks in Italy are mainly driven by pressure lows south of the peninsula, causing counterclockwise northward pulling of African desert air masses. While typically keeping its meridional character, this synoptic condition has been found to shift seasonally from the eastern in the spring to the western northern African coastal boundaries in the summer (Brattich et al., 2015a). Apart from regional-scale mechanisms, the transport of Saharan dust to the Mediterranean is essentially driven by large-scale dynamics and particularly by changes in atmospheric circulation patterns such as the North Atlantic Oscillation (NAO) (e.g., (Kaskaoutis et al., 2019) and references therein).

Differently from the vast literature on Saharan dust, the present work examines an outstanding mineral dust event whose characteristics are unprecedented to the best of our knowledge. The event described, indeed, concerns a singular and massive incursion of mineral dust reaching Europe from central Asia in late March 2020, a period marked by a series of peculiarities, not necessarily connected, such as an extensive, persistent, and intense eastern circulation, fairly infrequent at these latitudes, owing to basic dynamic constraints, the SARS COV-2 lockdown, and the concurrent, though seemingly out of context, large ozone hole in the Arctic (Manney et al., 2020). In particular, the trajectory of the dust plume was fairly unusual, since, as reported previously, dust storms from central Asia are usually transported to the south-west over the eastern part of Iran, Afghanistan or towards east-southeast, i.e., Pamirs, Tian-Shan mountains, and western China (Li et al., 2019; Zhang et al., 2020 and references therein). Furthermore, dust activity in central Asia is directly and indirectly influenced by atmospheric dynamics and teleconnection patterns such as the El Niño-Southern Oscillation (ENSO), the Arctic Oscillation (AO)/NAO, and the Siberian High (Zhang et al., 2020 and references therein).

The mineral dust incursion originating from the Caspian region widely bounced across the Italian media, as it led to $PM_{10}$ concentration levels well above the EU air quality threshold, wedging over an area historically characterized by frequent and significant exceedances of this pollutant such as the Po valley (Tositti et al., 2014), mainly due to the accumulation of anthropogenic emissions. As a result, this contingency led to harsh, though controversial, connections between air quality across the regional airshed and the virus spread (Belosi et al., 2021; Prather et al., 2020).

Differently from the extensive literature on Saharan dust, as previously mentioned, mineral dust from the Caspian region is far less investigated, though it is recently drawing ever-increasing attention by researchers owing to a multitude of climatic and environmental implications (Li et al., 2021). This arid region is indeed an area of increasing desertification owing to climate change as well as to decades of disastrous environmental management, including the desiccation of the Aral Sea, with a loss of 90% of its original water volume in the last decades (Behzod et al., 2012; Breckle and Wucherer, 2012; Loodin, 2020; Sharma et al., 2018; Shen et al., 2016; Shi et al., 2014; Zhang et al., 2020).

The event of mineral dust incursion from Central Asia herein described occurred at the end of March 2020 and, to the best of our knowledge, has not been so far the object of an in-depth analysis, though it was mentioned in a few papers devoted to other topics (Masic et al., 2020; Šikoparija, 2020) or described qualitatively in remote sensing or meteorological web pages (Mahovic et al., 2020; SNPA, 2020).

In this work, we present an in-depth analysis of this outstanding event based on the collection of high-resolution Optical Particle Counter number size distribution observed at three sites of NE Italy, i.e., at the two urban observatories in Bologna and Trieste, and the high altitude WMO-GAW (World Meteorological Organization-Global Atmosphere Watch) Global observatory of Mt. Cimone (https://cimone.isac.cnr.it/). Data analysis includes an accurate meteorological assessment in order to identify the peculiar synoptic conditions leading to the event. Back-trajectories analysis was used to identify the source region. Aerosol behavior during the dust incursion over Italy was analyzed by Optical particle spectrometry, while multiple data based on reanalysis (CAMS) and remote sensing (LIDAR and AERONET sun photometers) were employed to gain a dynamical 3D characterization of the mineral dust transport.

After the Introduction section and a description of the measurement techniques, the discussion of the results is organized as follows: 1) In-depth synoptic analysis and directional analysis of the event; 2) Analysis of aerosol size distribution in Bologna, Trieste, and Mt. Cimone before, during and after the event; 3) Influence on aerosol mass load 4) Trend of AOD, vertical distribution and chemical composition of the dust event. Finally, the main conclusions are summarized.

**2 Materials and methods**

**2.1 Sampling sites**

The Caspian dust event was investigated during a long-term air-quality campaign by three Optical Particle Counters (OPC, see paragraph 2.2 for details) located in three different sampling sites. Details of the three instruments are summarized in Table 1.

**Table 1.** Technical specifications of the three OPCs used: LOAC from MeteoModem, Multichannel Monitor from FAI Instruments, and GRIMM 1.108 from AMOF.

| Instrument | LOAC | FAI | GRIMM |
|---|---|---|---|
| Position | Bologna (44°29'N, 11°21'E) | Trieste (45°37'N, 13°46''E) | Mt. Cimone (44°11'N, 10°42'E) |
| Elevation (m a.s.l.) | 62 | 30 | 2165 |
| Dimensions (cm) | 20x10x5 | 45x44x17 | 24x24x6 |
| Weight (Kg) | 0.3 | 10 | 2.4 |
| Size range (μm) | 0.2-50 | 0.28-10 | 0.3-20 |
| Size bins | 19 | 8 | 15 |
| Flow rate (L min$^{-1}$) | ≈ 2 | 1.0 | 1.2 |
| Measurement frequency (s) | 1-60 | 60 | 6 |
| Laser wavelength (nm) | 650 | 630 | 655 |
| Scattering angle (°) | 12 and 60 | 90 | 90 |

In Bologna (BO), sampling was carried out with an OPC called LOAC (Light Optical Aerosol Counter; MeteoModem, 77760 Ury, France; size range 0.2-50 μm) (Brattich et al., 2020b; Renard et al., 2016b, 2016c). The instrument was installed on the rooftop of the Department of Physics and Astronomy of the University of Bologna (44°29'58''N, 11°21'14''E, 62 m a.s.l.). In addition, meteorological conditions (temperature, pressure, relative humidity, rain rate, wind speed, and direction) in Bologna were also collected by a Davis Vantage Pro2 weather station (Davis Instruments, Hayward, CA 94545, USA) on a 10-min time basis, located close to LOAC.

In Trieste (TS), an OPC Multichannel Monitor (FAI Instruments S.r.l., Roma, Italy; size range 0.28-10 μm) (Dinoi et al., 2017) was placed in the air quality station of via Pitacco (45°37'29''N, 13°46'46''E, 30m a.s.l.), managed by the Regional Environmental Protection Agency (ARPA FVG - Agenzia Regionale per la Protezione dell'Ambiente del Friuli Venezia Giulia). Meteorological data for Trieste are available as open data on the ARPA FVG website (https://www.osmer.fvg.it/) on a 1-h time basis.

At the summit of Mt. Cimone (CMN, 44°11'37''N, 10°42'02''E, 2165 m a.s.l.), is located the only WMO-GAW global station in Italy and within the Mediterranean basin. The atmospheric measurements carried out

at CMN can be considered representative of baseline conditions of the Mediterranean basin free troposphere. A GRIMM 1.108 OPC (AMOF, Leeds, UK; size range 0.3-20 μm) has been continuously running since August 2002 to observe accumulation and coarse mode aerosol number size distribution. The OPC is placed on a TSP (Total Suspended Particles) heated air inlet designed in the framework of EUSAAR (European Supersites for Atmospheric Aerosol Research) project and following the ACTRIS (Aerosol, Clouds and Trace gases Research InfraStructure) recommendation for aerosol inlets.

For all sites, data analysis for the present work covers the period between 20 March and 5 April 2020, namely from a week before the beginning of the dust transport event to northern Italy (27 March) and until one week after the end of the event (31 March).

## 2.2 Optical Particle Counters (OPCs)

Optical particle counters (OPCs) are widely used for aerosol characterization (Brattich et al., 2015a, 2019, 2020b; Bulot et al., 2019; Kim et al., 2019). The advantages of this class of aerosol instrumentation over the traditional filter-based instruments are their portability, the relatively low cost, the availability of aerosol size distribution, and the possibility of acquiring PM data continuously and at high time resolution (down to 1 measurement $s^{-1}$) (Brattich et al., 2020b).

OPCs exploit the principle of light scattering from aerosol particles, generally using a monochromatic high energy source such as a laser beam to detect and count particles. According to Mie scattering theory (Mie, 1908; Renard et al., 2016a), the intensity of scattered light is related to the particles' size, while the number of pulses of scattered light reaching the detector is related to the number of particles. This kind of instrument can retrieve a semi-continuous real-time analysis of the suspended particulate matter as a function of its diameter. In particular, the LOAC evaluates the scattering at two angles, 12° and 60°, the first being almost insensitive to the particles' refractive index, whereas the second is strongly sensitive to the refractive index (Renard et al., 2016b). FAI and GRIMM spectrometers detect aerosol particles from the scattered signal at 90°. All the OPC sensors operate at a 1 L $min^{-1}$ air volume flow rate and with a scan frequency of 1 min. For the sake of homogeneity, all the data were averaged at 1-hour time resolution. All OPC data are reported as number concentration, in counts $dm^{-3}$ (# $L^{-1}$).

The LOAC aerosol range is between 0.20 and 50 μm, while the FAI working range is between 0.30 and 10 μm and the GRIMM working range lies between 0.3 and 20 μm. Particle size distribution is obtained over 19 size bins for LOAC, 8 size bins for the FAI instrument, and 15 size bins for the GRIMM instrument, as reported in Table 2. A mean bin diameter was calculated for each size interval as indicated by Eq. 1 (Crilley et al., 2018):

$$D = LB \left[ \frac{1}{4} \left( 1 + \left( \frac{UB}{LB} \right)^2 \right) \left( 1 + \frac{UB}{LB} \right) \right]^{1/3} \tag{1}$$

where *LB* and *UB* are respectively the lower and the upper bin boundaries. While LOAC and GRIMM have

180 well-defined minimum and maximum particle diameters for each bin, FAI OPC works in an integral mode: the count value of each bin was obtained by subtracting the value of the following one and for the computation of *D*, *LB* of the latter bin was set as *UB*.

**Table 2.** Size bins of the three OPCs. All reported values are in μm

| Bin n° | LOAC OPC Boundaries | Mean diameter | FAI OPC Boundaries | Mean diameter | GRIMM OPC Boundaries | Mean diameter |
|---|---|---|---|---|---|---|
| 1 | 0.2 ÷ 0.3 | 0.253 | > 0.28 | 0.343 | 0.3 ÷ 0.4 | 0.352 |
| 2 | 0.3 ÷ 0.4 | 0.352 | > 0.4 | 0.452 | 0.4 ÷ 0.5 | 0.452 |
| 3 | 0.4 ÷ 0.5 | 0.452 | > 0.5 | 0.606 | 0.5 ÷ 0.65 | 0.578 |
| 4 | 0.5 ÷ 0.6 | 0.551 | > 0.7 | 0.915 | 0.65 ÷ 0.8 | 0.728 |
| 5 | 0.6 ÷ 0.7 | 0.651 | > 1.1 | 1.59 | 0.8 ÷ 1.0 | 0.904 |
| 6 | 0.7 ÷ 0.9 | 0.804 | > 2.0 | 2.53 | 1.0 ÷ 1.6 | 1.32 |
| 7 | 0.9 ÷ 1.1 | 1.00 | > 3.0 | 4.08 | 1.6 ÷ 2.0 | 1.81 |
| 8 | 1.1 ÷ 3.0 | 2.19 | > 5.0 ÷ 10.0 | 7.77 | 2.0 ÷ 3.0 | 2.53 |
| 9 | 3.0 ÷ 5.0 | 4.08 | | | 3.0 ÷ 4.0 | 3.52 |
| 10 | 5.0 ÷ 7.5 | 6.33 | | | 4.0 ÷ 5.0 | 4.52 |
| 11 | 7.5 ÷ 10.0 | 8.81 | | | 5.0 ÷ 7.5 | 6.33 |
| 12 | 10.0 ÷ 12.5 | 11.3 | | | 7.5 ÷ 10.0 | 8.81 |
| 13 | 12.5 ÷ 15.0 | 13.8 | | | 10.0 ÷ 15.0 | 12.7 |
| 14 | 15.0 ÷ 17.5 | 16.3 | | | 15.0 ÷ 20.0 | 17.6 |
| 15 | 17.5 ÷ 20.0 | 18.8 | | | > 20.0 | |
| 16 | 20.0 ÷ 22.0 | 21.0 | | | | |
| 17 | 22.0 ÷ 30.0 | 26.2 | | | | |
| 18 | 30.0 ÷ 40.0 | 35.2 | | | | |
| 19 | 40.0 ÷ 50.0 | 45.2 | | | | |

Owing to the different ranges and number of bins of the used OPCs (Table 2), experimental data were homogenized as follows. Seven aerosol fractions were considered for comparison of the three OPC outputs, while two additional fractions were evaluated for LOAC only: "fr0.3", "fr0.4", "fr0.5", "fr0.7", "fr1.1", "fr3", and "fr5" for all OPCs; "fr0.2" and "fr10" for LOAC only. Table 3 describes the processed fractions and the bin combination used for the comparison. Table 2 shows that also the GRIMM instrument could be

used to evaluate "fr10". However, previous studies (Duchi et al., 2016; Marinoni et al., 2008; Sajani et al., 2012) highlighted that the presence of coarse particles is unusual on the top of CMN, except in cases of Saharan dust (Marinoni et al., 2008). Moreover, as described further on, the event herein analyzed, had limited vertical development and did not significantly affect the top of Mt. Cimone, as shown by the low counts of the coarser bins at this station. Therefore, we decided not to consider "fr10" for CMN.

**Table 3.** Fractions analyzed with LOAC, FAI OPC, and GRIMM, with the corresponding bin combinations (sums or differences)

| Fraction | Size interval (μm) | LOAC | FAI | GRIMM |
|---|---|---|---|---|
| fr0.2 | 0.2 ÷ 0.3 | bin1 | -- | -- |
| fr0.3 | 0.3 ÷ 0.4 | bin2 | bin1 – bin2 | bin1 |
| fr0.4 | 0.4 ÷ 0.5 | bin3 | bin2 – bin3 | bin2 |
| fr0.5 | 0.5 ÷ 0.7 | bin4 + bin5 | bin3 – bin4 | bin3 |
| fr0.7 | 0.7 ÷ 1.1 | bin6 + bin7 | bin4 – bin5 | bin4 + bin5 |
| fr1.1 | 1.1 ÷ 3.0 | bin8 | bin5 – bin7 | bin6 + bin7 + bin8 |
| fr3 | 3.0 ÷ 5.0 | bin9 | bin7 – bin8 | bin9 + bin10 |
| fr5 | 5.0 ÷ 10.0 | bin10 + bin11 | bin8 | bin11 + bin 12 |
| fr10 | > 10.0 | Sum from bin11 to bin 19 | -- | -- |

Computations and graphs of time-series, particle size distributions, and number concentration to mass conversions were performed with R software (R Core Team, Vienna, Austria). Polar plots of the OPC fractions were calculated considering all data from 20 March to 6 April with the R package "*openair*" (Carslaw and Beevers, 2013; Carslaw and Ropkins, 2012).

**2.3 Synoptic-scale conditions**

The synoptic-scale main patterns related to the dust transport have been investigated based on the meteorological fields provided by the Global Forecast System (GFS) coupled model, produced by the National Centers for the Environmental Prediction (NCEP). In particular, we have analyzed the pressure field at the sea level (SLP) and the geopotential height at 500 (Z500) and 850 (Z850) hPa at 00:00 UTC for each day during the period of the event.

Besides, to locate and follow such long-range transport, we have used the regional air quality products of the Copernicus Atmosphere Monitoring Service (CAMS) (Marécal et al., 2015) in terms of particulate matter concentration (specifically $PM_{10}$ fraction) at the height of 50 m above the surface. In detail, we have analyzed the output provided by the CAMS that merges model and observation data and ensures a high spatial resolution (~0.1 degrees) (CAMS, 2015).

Back-trajectories were also evaluated to confirm the origin of the studied mineral dust event from Central Asia. Back-trajectories are computed simulating the transport of air masses for a determined time frame until these reach a given "receptor" point at a specific time instant (Fleming et al., 2012; Rolph et al., 2017). In the specific case, 96-h back-trajectories were computed by the Hybrid Single-Particle Lagrangian Integrated Trajectory (HYSPLIT_4) model (Rolph et al., 2017; Stein et al., 2015) considering as receptor sites the three locations of the OPCs (see paragraph 2.1) over the whole period of the event. The computation was performed every six hours starting from 27 March 00:00 UTC until 31 March 18:00 UTC using GFS meteorological data at a 0.25-degree (27.8 km) resolution (National Centers for Environmental Prediction/National Weather Service/NOAA/US Department of Commerce, 2015). For each receptor site, trajectories were computed at different heights to determine the impact of the arrival height on the trajectory analysis: 100, 1000, and 2000 m a.g.l. for Bologna (BO) and Trieste (TS), and 1700, 2200, 2700, and 3200 m a.g.l. for Mt. Cimone (CMN).

### 2.4 Particulate matter concentrations and chemical speciation data

In addition to measurements from the OPCs, daily mean $PM_{10}$ concentrations measured with automatic low-volume SWAM 5a Dual Channel Monitor (Fai Instruments, Rome, Italy) at two air quality stations of the ARPAE (Agenzia Prevenzione Ambiente Energia Emilia Romagna) Regional Environmental Protection Agency in Bologna (Porta San Felice and Giardini Margherita, respectively urban traffic and urban background stations) were used. Data were obtained through the "*saqgetr*" R package (Grange, 2019).

Furthermore, chemical speciation data (ions, carbonaceous fraction, and elements) sampled in Bologna and available as open data from ARPAE ([https://www.arpae.it/it](https://www.arpae.it/it)) were also analyzed to complement the other analyses. The two mentioned air quality stations are located 2 km from the LOAC site, to the west and the south, respectively. The data are validated daily, monthly, and every six months and are certified according to UNI EN ISO 9001:2015 standard.

### 2.5 AERONET (AErosol RObotic NETwork)

Aerosol optical depth (AOD) daily averages data retrieved from Aerosol Robotic Network (AERONET) ground-based remote sensing aerosol network at the site of Venice (Acqua Alta Oceanographic Tower - AAOT; 45°19'N, 12°30'E) were used as an independent method to confirm the dust aerosol transport event at the end of March 2020.

AERONET collaboration provides globally distributed observations of spectral aerosol optical depth (AOD), inversion products, and precipitable water in diverse aerosol regimes. For this work, version 3 AOD data and inversions computed for Level 2.0, i.e., quality assured (with pre- and post-field calibration applied, automatically cloud cleared, and manually inspected), were utilized. Total mode, fine mode, and coarse mode AOD at 500 nm (standard reference wavelength) were computed using a best-fit second-order polynomial according to the spectral deconvolution algorithm (SDA) by O'Neill et al. (O'Neill et al., 2003). Finally, the AERONET inversion code provides aerosol optical properties in the total atmospheric column derived from the direct and diffuse radiation measured by AERONET Cimel sun/sky-radiometers (NASA, 2006). The output includes both retrieved aerosol parameters, which comprehends the volume size distribution, and parameters calculated based on the retrieved aerosol properties. The inversion algorithms are based on the following assumptions, i.e.: the atmosphere is considered as plane-parallel; the vertical distribution of aerosol is assumed homogeneous in the almucantar inversion and bi-layered for the principal plane inversion; aerosol particles are assumed to be partitioned in spherical and non-spherical components.

### 2.6 LIDAR Ceilometer

Aerosol vertical profiles collected in Milan (Diémoz et al., 2019; Ferrero et al., 2020) (45°31'N 9°12'E, 130 m a.s.l., ~200 km northwest of Bologna) have been used to identify the signature of dust transport as well as to investigate its vertical extension and altitude. Such profiles were gathered by a Nimbus CHM15K system

(Lufft, Germany) running within the Italian Automated Lidar Ceilometer network – ALICENET (www.alice-net.eu), coordinated by ISAC-CNR in partnership with other Italian research institutions and environmental agencies. It is a high-performance system providing vertical profiles of aerosols and clouds in the first 15 km of the atmosphere with a temporal resolution of 30 s and a vertical resolution of 15 m (Dionisi et al., 2018). ALICENET measurements have been and are usefully employed to detect altitude and temporal evolution of cloud layers (Ferrero et al., 2020) and to track transport of polluted (Diémoz et al., 2019) or mineral dust aerosol plumes (Gobbi et al., 2019) at different sites along the Italian peninsula.

# 3 Results and discussion

## 3.1 Synoptic analysis

As documented later on, the source region of the investigated episode of severe dust transport was the Central Asian southern desert region located between the eastern coast of the Caspian Sea and the steppes near the central Asia mountain ranges. Most of Turkmenistan and eastern Uzbekistan, as well as the Central Asian desert ecoregion to the North, can be considered a cold desert, in agreement with the inherent climatic class (BWk, according to the Köppen classification), with hot summers and cold winters, and annual precipitation of 125–170 mm $year^{-1}$, with winter and spring as the wettest seasons (Li et al., 2019, 2021). In order to analyze the weather patterns during the event, we have decided to analyze the pressure and wind fields at the sea level (SLP) and the geopotential height at 500 (Z500) and 850 (Z850) hPa in the period from 20 March to 30 March 2020. Figure 1 shows the geopotential maps Z850, while Fig. 2 shows the corresponding $PM_{10}$ (50 m height) concentration maps and the surface horizontal wind derived from CAMS reanalysis for the period 24 – 31 March 2020, which can be considered the period in which the dust transport took place. Figure A1 in the appendix, instead, shows the same geopotential maps Z500.

On 20 March, at 500 hPa, the geopotential height was high on the western Mediterranean basin and western Europe due to a ridge expanded from Libya to Italy and Poland, while the pressure field at the sea level and the geopotential height at 850 hPa were uniform. On the Caspian deserts, the circulation was weak, even if, in the northern area, southern winds were flowing towards Uzbekistan and Kazakhstan, and the 850 hPa winds were westwards at the East of the Caspian Sea. On this date, there were three areas showing a large quantity of dust concentration: northern Africa, western Kazakhstan and Turkmenistan, and Iraq and Syria (originated from the Arabic peninsula).

In the following three days, regarding Z500, the European anticyclone elongated versus northeast reaching Scandinavia on 23 March, and at the same time, a trough developed from Russia extending to Bosnia and Albania; regarding SLP and Z850, an anticyclone developed on southern Sweden, activating a weak eastern flow directed from the southern Urals to Croatia. Over the Caspian desert, there was a complete change of circulation, and winds turned westwards during 22 March at the east of the Caspian Sea, then turning northwards over the Caspian Sea. At that moment, there was the onset of the dust transport (in the subsequent days) towards Europe and therefore Italy. Note that the characteristics of the southward extension of the Siberian high during these days, and the associated pressure gradients, were favorable to the development of dust storm over the Caspian region, as indicated by previous works (Hamidianpour et al., 2021; Labban et al., 2021; Li et al., 2019, 2021). On the central Mediterranean Sea, a strong southern flow from Libya, pivoted by a minimum between Tunisia and Sicily, established, bringing Saharan dust over the Ionian Sea and Sicily on 22 March and over Greece and western Turkey on 23 March.

On 24 March (Fig. 1a, 2a, and A1a), at Z500, a narrow ridge was extending from the Azores to Baltic Republics, while a minimum formed between Caspian deserts and Italy; regarding SLP and Z850, a strong anticyclone was present over Lithuania and Belarus; a minimum north of the Caspian Sea originated an intense eastern flow from the Caspian desert to the eastern Black Sea. Another southward flow was present over eastern Europe, directed over the western Black Sea, which subsequently turned right, becoming westward and moved towards Italy. At that time, the Italian Adriatic regions were already affected by a strong NE flow, but without dust transport, since the Caspian desert dust cloud was just approaching the eastern Black Sea on the evening of that day. Simultaneously, due to the cyclonic curvature of the flow on the eastern Mediterranean Sea, some dust started to move from Syria to Turkey, while Saharan dust flow towards Italy was temporarily paused.

On 25 March (Fig. 1b, 2b, and A1b), at Z500, a cutoff was gradually filling over Italy, surrounded by a ridge from the UK to Belarus and another one West of Urals, in Asia. Regarding SLP, isobars outlined an almost linear corridor of flow between the Caspian deserts and Bosnia and Croatia, which veered North across the Bora door (a section of the Alps through which the Bora wind reaches Italy from the Balkan region) entering in northeast Italy; regarding Z850, a remarkable flow was present between Romania and Croatia. The Caspian desert dust cloud crossed the Black Sea during this day, still traveling near the surface. Important for the next evolution was also the minimum present in the SLP and at Z850 over Tunisia, which was responsible for a new Saharan dust transport towards the Ionian Sea and Sicily. Meanwhile, the Syrian dust continued moving westwards, reaching Bulgaria, probably mixed with Saharan dust that was moving northwards from Egypt.

On 26 March (Fig. 1c, 2c, and A1c), the previously mentioned anticyclone extended towards the Black Sea, joining with the one over Egypt, while the cutoff retreated to Italy and the western Mediterranean Sea, promoting the shift of the SLP and Z850 anticyclone to the north of the Black Sea. The minimum over Tunisia shifted to the Ionian Sea, thus strengthening the flow along a corridor extended from the Caspian desert across the Black Sea up to Romania and Bulgaria and then to the Adriatic Sea. Under these conditions, the most intense flow was observed from the Black Sea to the Adriatic Sea. In the evening, the Aralkum desert dust cloud reached Croatia and the Adriatic Sea, where it merged with the Sirocco wind associated with the Ionian Sea cyclone, responsible for Saharan dust transport towards southern Italy and, in minimal part, Italian central Adriatic regions. Meanwhile, a strong south-eastern wind established on the eastern Mediterranean Sea, favoring dust transport from Iraq and Saudi Arabia across Syria up to western Turkey and Greece. At this stage, the area between Romania, Croatia, Greece, Albania, Bulgaria, and Turkey was affected simultaneously by dust coming from three different source areas ranging from northern Africa, the Arabic peninsula, and eastern Caspian deserts; this represents something really unusual.

On 27 March (Fig. 1d, 2d, and A1d), the trough at Z500 was located on the western Mediterranean basin, surrounded by anticyclones and ridges; SLP and Z850 cyclone moved over the Tyrrhenian Sea, while the anticyclone also expanded over Romania; as a consequence, the main flow from the Caspian deserts, moving

across the Black Sea, headed towards Greece, veering toward northwest, thereafter affecting the Adriatic Sea and eastern Italy. The Caspian desert dust cloud touched down on the coast of the Adriatic regions (Friuli Venezia Giulia, Emilia Romagna, and Marche) in the evening. Over the Aegean Sea, instead, there was dust that probably can be considered a mixture of two sources (Caspian deserts and Arabic peninsula), pushed westwards by the strong eastern flow present over eastern Europe and the eastern Mediterranean Sea.

On 28 March (Fig. 1e, 2e, and A1e), due to the weakening of the geopotential field at Z500 and Z850, as well as the SLP field, the flow from Romania to Italy considerably weakened. The circulation over northeastern Italy changed due to the influence of the anticyclonic curvature induced by a large high of SLP located between Iceland and Ireland, pivoting a strong surface northern flow from Scandinavia. A first impulse of cool easterly air masses shifted along the surface across the Po valley in the early morning, forming a shallow front (with few clouds) able to transport the dust cloud up to the border of the Piedmont region (northwestern Italy) during the afternoon. Over eastern Italy, instead, the local breeze circulation was prevailing.

During the evening, a second easterly impulse crossed the Po Valley, this time accompanied by more clouds but without precipitations, and during the morning of 29 March (Fig. 1f, 2f, and A1f) also reached the northwestern Italian regions (Piedmont, Valle d'Aosta, and Liguria). At that time, approximately half of Italy was affected by the Caspian desert dust cloud, though over the northeast sector, the concentration was already decreased owing to a northerly air mass from the Alps. On that date, SLP and geopotential at Z850 and Z500 abruptly decreased in a corridor from Baltic republics to Greece, thus completely interrupting the easterly flow of dusty air, which remained active just over the eastern Mediterranean basin. During the afternoon of 29 March, the inflow of the arctic air mass moving across the Alps displaced the mineral dust-rich air mass over the Po Valley southwards.

Finally, on 30 March (Fig. 1g, 2g, and A1g), the cold front pushed by the Arctic airflow fully embraced the Po valley, thus definitely cleaning most of northern Italy, as it was visible also on the morning of 31 March (Fig. 1h, 2h, and A1h).

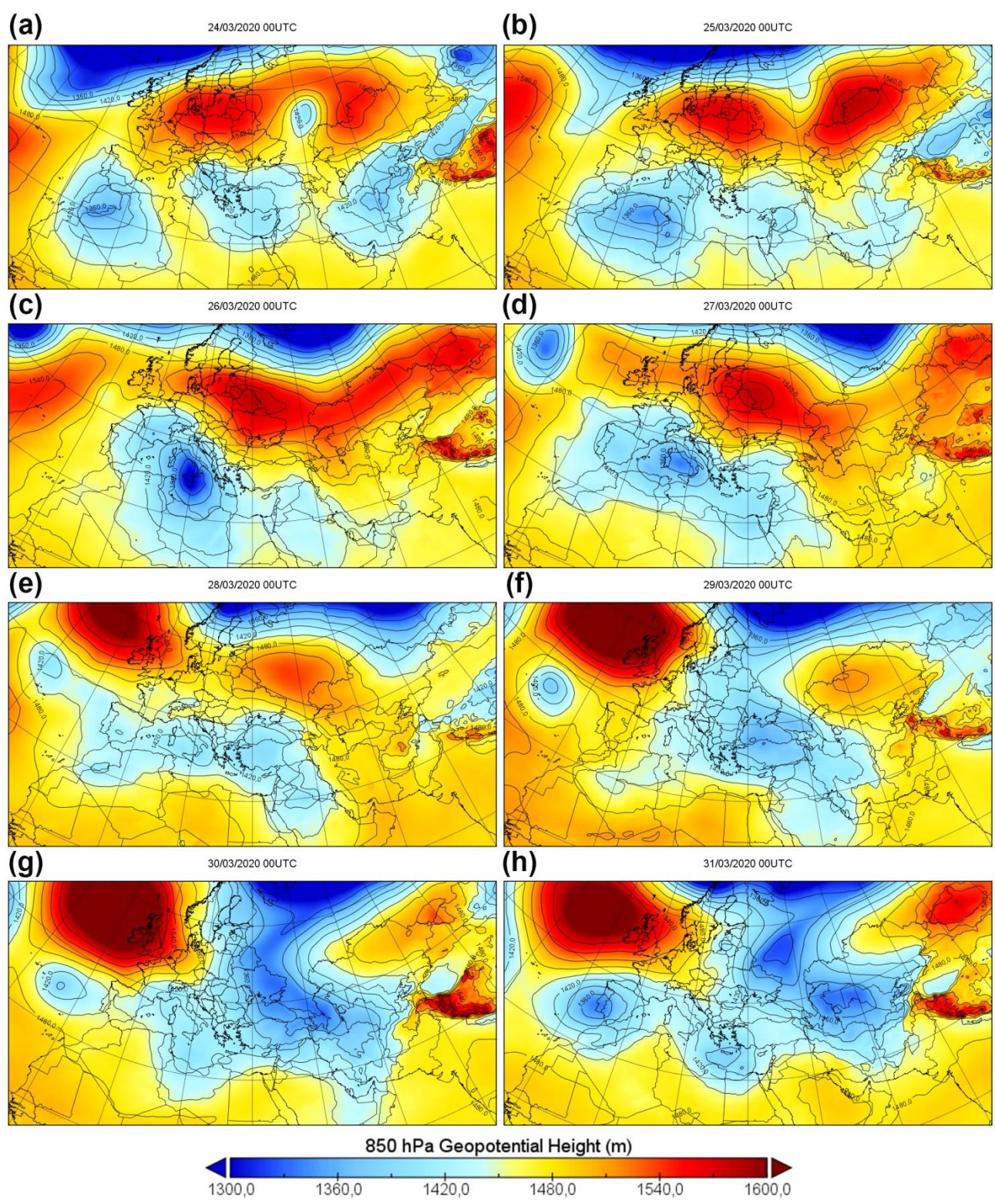

**Figure 1.** Geopotential at 850 hPa (in m$^2$ s$^{-2}$) maps (colors and isolines) relative to each day in the period 24-31 March at 00 UTC (a-h). The maps were created with Panoply starting from ERA5 reanalysis. Letters from a) to h) are referred to the days from 24 to 31 March 2020.

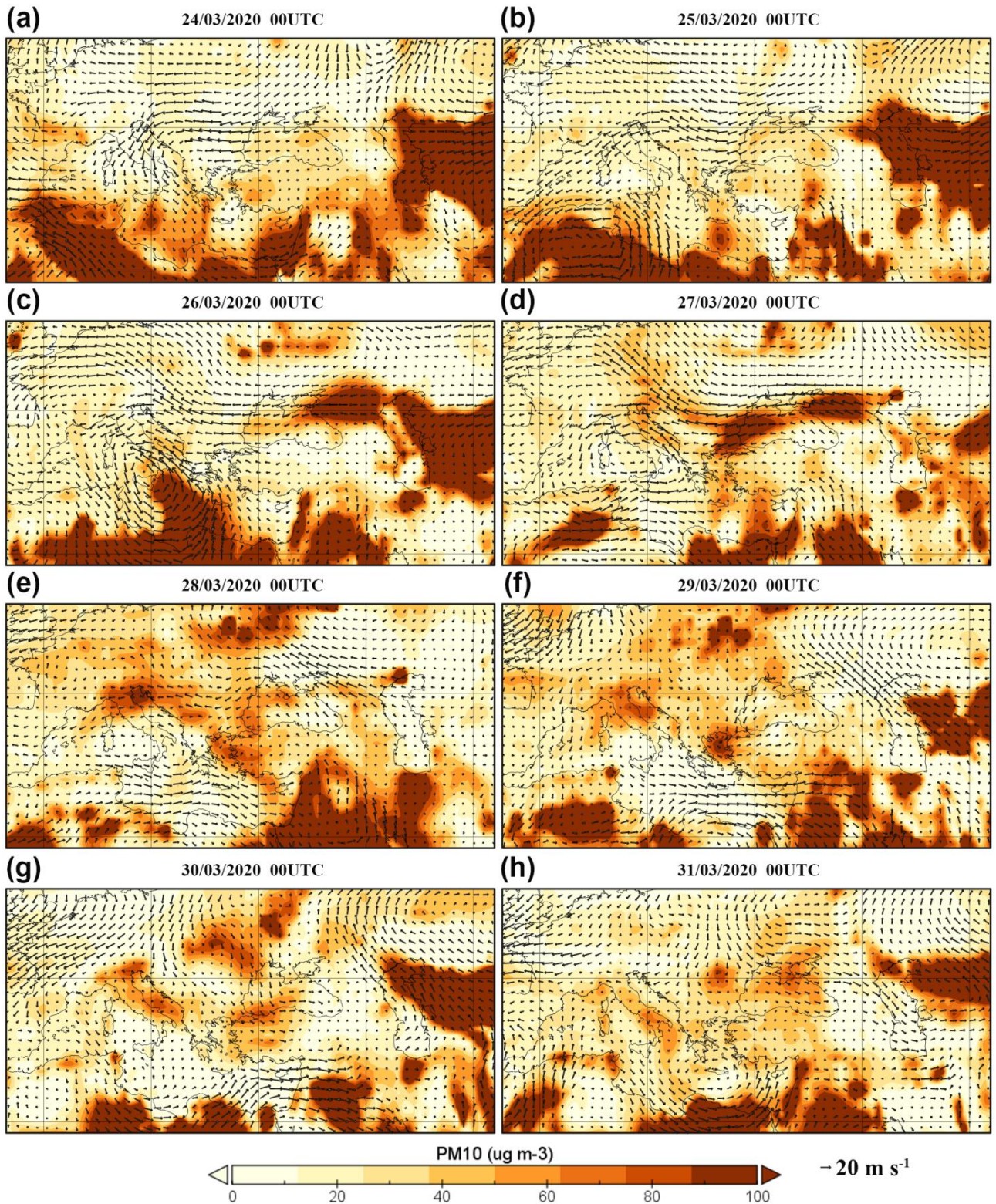

**Figure 2.** $PM_{10}$ (in μg m$^{-3}$) maps (colors) and horizontal surface wind (arrows) relative to each day in the period 24-31 March at 00 UTC (a-h). The maps were created with Panoply starting from ERA5 reanalysis. Letters from a) to h) are referred to the days from 24 to 31 March 2020.

The easterly origin of the air mass originating the dust event over northern Italy was also confirmed by back-trajectory analyses. 96-h back-trajectories were computed using the three measurement sites (BO, TS, and

370 CMN) as receptor points. The most relevant back-trajectories are reported in Fig. 3 (for BO) and in the Appendix A for TS (Fig. A2) and CMN (Fig. A3).

Fig. 3a shows, in particular, the back-trajectory ending on 27 March 18:00 UTC in BO. Such date corresponds to the first peak due to the Caspian dust observed in TS by FAI-OPC, considered as the beginning of the event in Italy. Fig. 3a already indicates an airflow originating between Caspian and Aral 375 Seas, with some intrusions from the Black and Mediterranean Seas. The Caspian origin of air masses becomes even more evident in the following days (Fig. 3b and 3c). Then (Fig. 3d), on 30 March, a northern stream started to flow over northern Italy, cleaning the air from the dust. The observation drawn from these back-trajectories confirms what was suggested by the synoptic analysis. The calculated trajectories, within the limits of the computational uncertainty, suggest that the Caspian dust reached Trieste passing through the 380 Bora door before arriving in Bologna,

This evidence is also confirmed by the computed back-trajectories for the TS measurement site shown in Fig. A2. The ending times in Fig. A2 are the same as Fig. 3, and the conclusions that can be drawn are similar: an airflow originating in the Caspian region reached Trieste on 27 March (Fig A2a), and it continued in the following days (Fig. A2b and A2c) until a northern stream cleaned the air starting from 30 March (Fig. A2d).

Figure A3 shows, instead, some of the calculated back-trajectories for Mt. Cimone. In this case, an air mass with a possible north-African origin, with a strong component from the Aegean Sea, reached the site on 27 March (Fig. A3a), although it did not give origin to visible peaks in the OPC time-series (Fig. 4c and A4c). Then, again (Fig. A3b and A3c), the peaks observed on 28 – 29 March can be associated with an airflow deriving from the Caspian region.

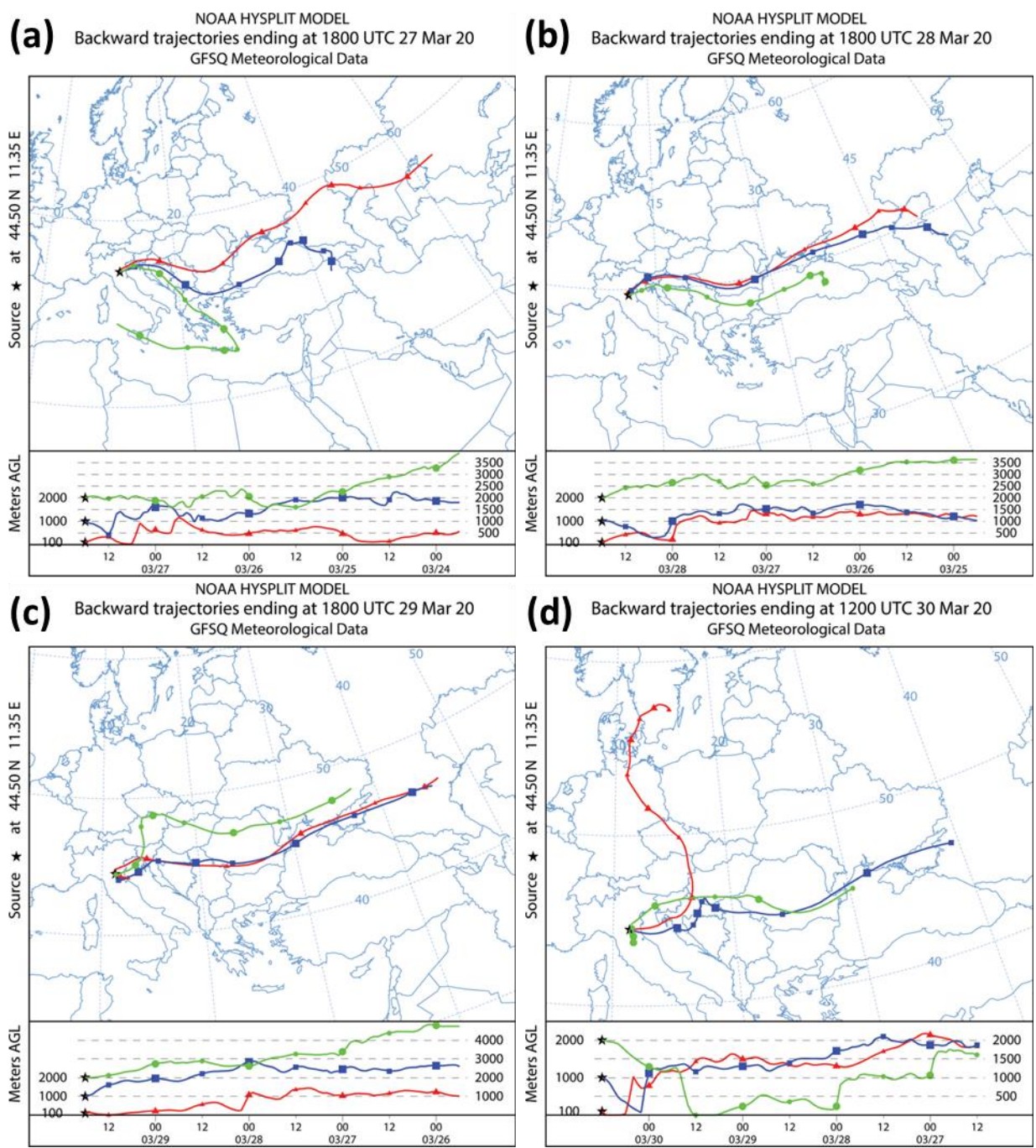

**Figure 3**. Back-trajectories (96-h backwards) ending at Bologna on a) 27 March 18:00 UTC; b) 28 March 18:00 UTC; c) 29 March 18:00 UTC; d) 30 March 12:00 UTC

**3.2 Optical Aerosol Counters (OPCs)**

**3.2.1 Temporal trends and particle size distributions**

The mineral dust outbreak produced a remarkable increase in particle number densities at all the three sites investigated, i.e., Bologna, Trieste, and at the top of Mt. Cimone. This increase was remarkably high for the coarse fractions, a typical feature of mineral dust, in particular at both the urban stations.

Figure 4 reports the temporal evolution of the coarse fractions limited, for the sake of simplicity, to fr5 (diameter 5-10 μm) for BO and TS, and the fraction fr3 (3-5 μm) for CMN. For the latter, fr3 was considered, instead of fr5, due to a large number of data below the detection limit for size bins > 7.5 microns, as previously reported in the Materials and methods section. This is consistent with a possible loss of larger particles due to gravitational settling during the transport/interactions at higher altitude (Mallios et al., 2020). The complete OPC series are available in Appendix A (Fig. A4).

As reported in Fig. 4, coarse particle number densities were very low before and after the event, reflecting the typical size distribution at urban locations in the cold season. Conversely, the fairly low level in the fine fraction (Fig. A4) is likely due (besides the typical seasonal pattern at this latitude characterized by a decrease in the warmer season  (Perrino et al., 2014)), at least in part, to the consequences of the lockdown imposed to stem the spread of the SARS-COV2 pandemic in Italy, from early March 2020 (Chauhan and Singh, 2020; Donzelli et al., 2020; Malpede and Percoco, 2021). The Asian dust plume impacted TS first, on 27 March during the afternoon, reaching BO (about 200 km south-west) the day after (please note the time lag of the order of one day at the two stations in Fig. 4a and 4b). Overall, the event duration was about three days, ceasing on 30 March in Trieste and 31 March in Bologna. During this period, a general increase in suspended particle concentration, especially in the coarse fraction, was observed at all the stations in agreement with mineral dust properties and its resulting influence on PM mass load recorded at the time throughout several European stations (Mahovic et al., 2020). The dust plume reached CMN (Fig. 4c) on 28 March, but the maximum concentrations were slightly lagged, peaking between the afternoon of 29 March until 30 March. Another peak in the CMN series is visible on 21 March, likely due to a genuine "high-pass" Saharan dust, which is not herein discussed being beyond the scope of the present work.

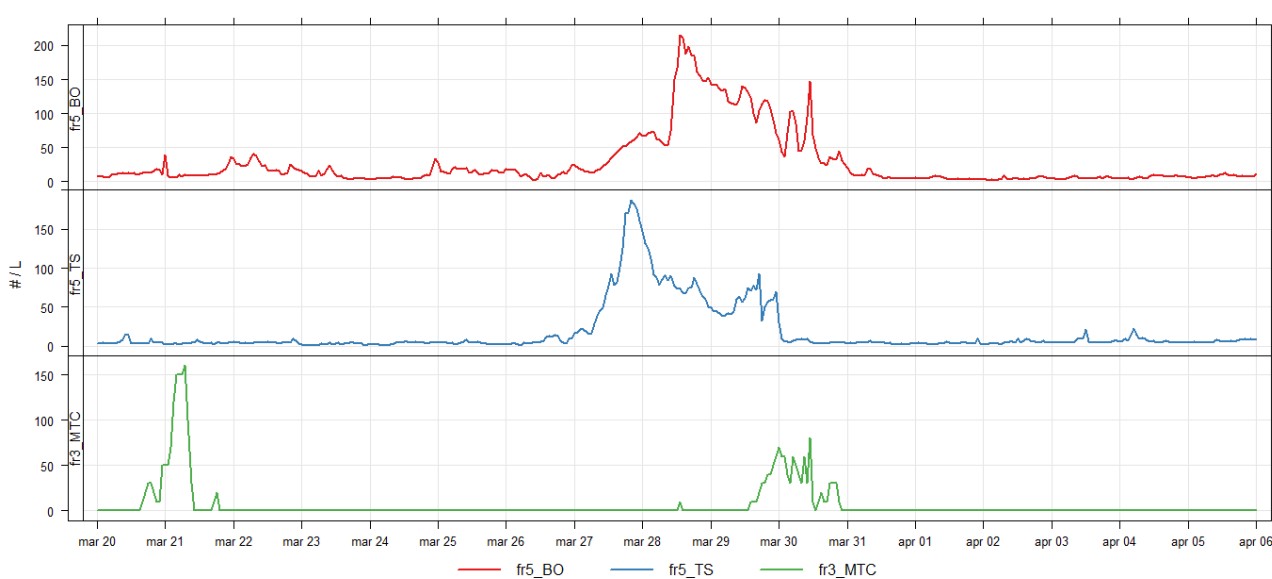

**Figure 4.** Temporal trends of fr5 (5.0 to 10.0 μm) at Bologna, red line, and Trieste, blue line, and of fr3 (3.0 to 5.0 μm) at Mt. Cimone, green line. Values are in Counts dm$^{-3}$ (# L$^{-1}$)

OPC data were associated with the meteorological parameters collected at the three sites. In particular, the connection with local wind speed and direction was evaluated using polar plots (Carslaw and Beevers, 2013).

For the present study, we computed polar plots for fr5 of both cities and for fr3 of CMN as reported in Fig. 5. A conditional probability function (CPF) at the 90[th] percentile was used for computation in order to minimize pollution source effects and focus the computation only on the event of dust transport (Kurniawati et al., 2019). Polar plots of both cities (Fig. 5a and 5c) show a maximum CPF probability corresponding to the wind blowing from the east direction, agreeing with the advection of dust from the Caspian region to the

study area. The wind speed corresponding to the two wind intensity maxima was about 3 m s$^{-1}$ for Bologna and 7 m s$^{-1}$ for Trieste. The polar plot of CMN (Fig. 5e) shows, instead, a probability maximum for slow winds (3-5 m s$^{-1}$) coming from the south-west, corresponding to 30 and 31 March (Fig. 4 and 5f). Such maximum is mainly due to the low intensity of wind rather than to its direction. Indeed, the previous days are characterized by a strong northeasterly wind (compatible with the dust origin area), even stronger than 25 m

s$^{-1}$, a situation under which the OPC was not able to register any increase in particles. Also in TS, the transport of the dust plume was preceded by a strong easterly wind, about 15 up to 20 m s$^{-1}$ (Fig. 5d), while during the event, the wind intensity decreased to 2-5 m s$^{-1}$. In this case, another strong easterly current coming from the Bora door cleaned the air and ended the event. BO wind (Fig. 5b), instead, did not show a clear pattern. The weak winds registered in Bologna, even lower than 1 m s$^{-1}$ during the event, are typical of

the Po Valley basin. The Easterly direction remains the most remarkable feature since this source area is fairly infrequent for air masses arriving in Italy, though acknowledged especially in the spring (Battiston et al., 1988; Brattich et al., 2015b, 2020a; Dimitriou and Kassomenos, 2014).

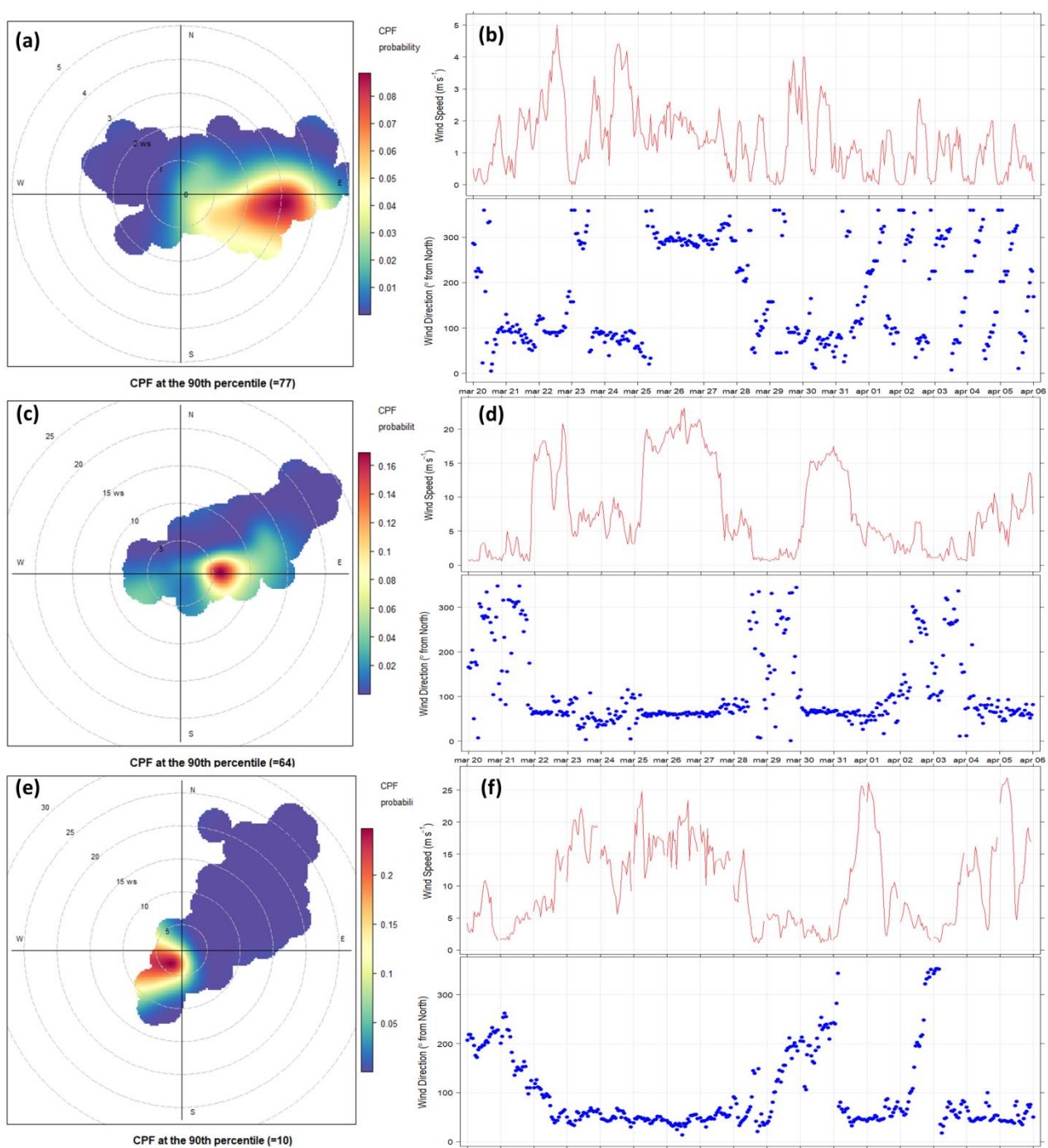

**Figure 5.** Right part: polar plots of a) fr5 for Bologna; c) fr5 for Trieste; e) fr3 for Mt. Cimone. Gray dashed circumferences indicate wind speed in m s$^{-1}$. Left part: wind speed (red line) and wind direction (blue points) from 20 March to 6 April 2020 for b) Bologna; d) Trieste; f) Mt. Cimone

The temporal behavior of particle number densities as a function of size is not the same (Fig. A4). At the three stations, mineral dust transport showed a stronger increase for the coarse fraction, while the increase in fine particles, although present, is generally slighter. Mt. Cimone, instead, showed an apparent strong increase in the fine fractions, but not in the coarse ones suggesting a depletion due to the influence of gravitational settling on particles with height.

This information can be summarized by comparing particle size distributions at the three stations before (black line), during (blue line), and after the event (pink line), as reported in Fig. 6. An overall increase in number densities is observed at all sites during the dust event. However, as reported in Fig. 6, the most significant deviations of the within-the-event period from both before- and after-event periods were observed starting from 0.7 μm in BO and TS, while at CMN, also the finest particles undergo an increase during the event. This confirms the primary influence of the dust on coarse particles, rather than on finer ones, at least in the two cities. This finding is different from Saharan dust outbreaks since, in this latter case, also the finest fractions increase, at least among the size range determined by OPC and/or from LIDAR observations (Brattich et al., 2015a; Denjean et al., 2016).

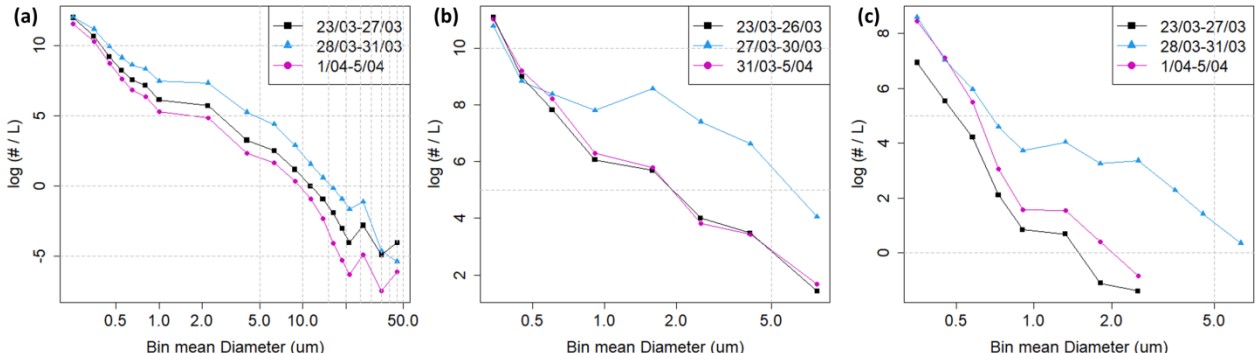

**Figure 6.** Particle size distributions for the OPCs: a) LOAC (Bologna); b) FAI (Trieste); c) GRIMM (Mt. Cimone). Abscissa values are in μm, ordinate values in log(# L$^{-1}$). Abscissa axis is on logarithmic scale

### 3.2.2 Particle Mass Concentrations

Number densities were eventually converted into daily PM$_{10}$ mass loads (Fig. 7). Details of the algorithm used for such conversion are described elsewhere (Brattich et al., 2020b). It uses the lower and upper boundaries of each bin size to calculate a "mean" bin diameter (Eq. 1), which is then used to calculate PM concentrations. The last computation is based on an average PM density of 1.65 g cm$^{-3}$, usually applied to urban aerosol composition, which may not accurately fit mineral dust aerosols with a different chemical composition (Brattich et al., 2020b; Gholamzade Ledari et al., 2020). Calculated daily PM$_{10}$ mean in BO reached values of 79 and 74 μg m$^{-3}$ respectively on 28 and 29 March (with an absolute maximum of 168 μg m$^{-3}$ on 28 March 13:00 UTC), with a negative bias of about 12-15% with respect to the experimental values recorded by the Regional Environmental Protection Agency in Bologna, revealing an expected underestimate (Fig. 7a). Calculated PM$_{10}$ values were even higher at TS, where 107 and 124 μg m$^{-3}$ were reached on 27 and 28 March (with maxima of 275 at 19:00 UTC and 246 μg m$^{-3}$ at 23:00 UTC on 27 March). Recorded PM$_{10}$ values in Bologna and Trieste were remarkably higher than the EU PM$_{10}$ threshold (European Parliament, 2008) and 3-5 times the WHO recommendation (WHO for European, 2006) and the estimation of the desert dust load following the method suggested by Barnaba et al. (2017) indicates values in the range of 66-108 μg m$^{-3}$ at these two cities (Fig. 7b).

In BO, the OPC registered $PM_{10}$ slightly above the EU threshold also on 30 March (54 μg m$^{-3}$), while two relative maxima were detected on 27 March (36.4 μg m$^{-3}$) and 22 March (35.0 μg m$^{-3}$). The $PM_{10}$ mean of the days before the event (from 20 to 26 March, excluding 22 March), which can be considered as representative

of the typical background of the region likely due to the limited influence of vehicular emissions during the lockdown and the low influence of industrial activities in Bologna, were 15.0 μg m$^{-3}$ and 10.3 μg m$^{-3}$ respectively prior and after the mineral dust outbreak event. A tail of the event was also observed in TS on 29 March with 79.4 μg m$^{-3}$. The $PM_{10}$ daily mean value of the days before the event in TS was 11.4 μg m$^{-3}$, while that of the days immediately following the event (31 March – 6 April) was 13.9 μg m$^{-3}$.

An increase in calculated $PM_{10}$ was also observed at Mt. Cimone with a maximum of 11 μg m$^{-3}$ on 30 March 11:00 UTC. Although this value is very low compared to the ones of Bologna and Trieste, the increase is outstanding if compared with the mean $PM_{10}$ values observed before and after the event: 0.11 and 0.35 μg m$^{-3}$, respectively in agreement with winter relative minima previously reported, due to the height (2165 m a.s.l.) and the cold-season decoupling of the mountain top from the PBL (Tositti et al., 2013).

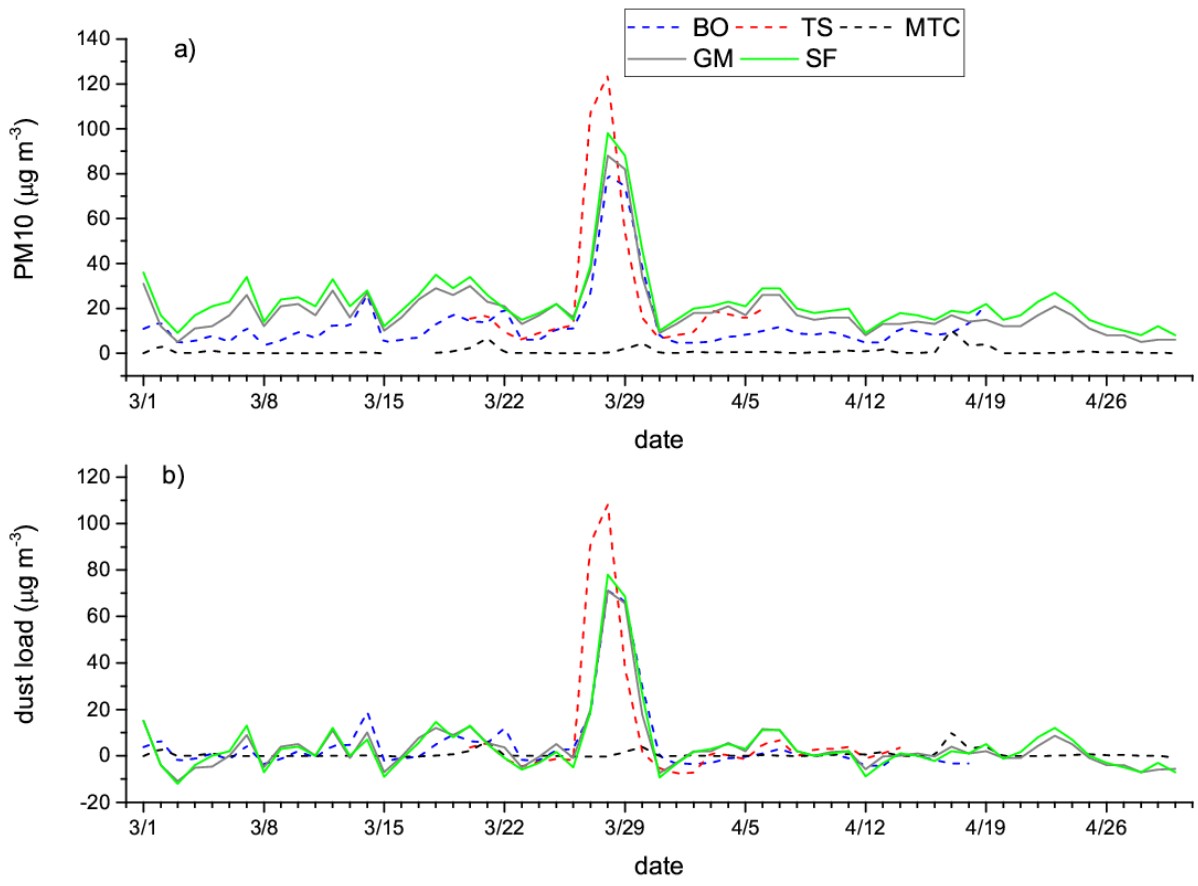

**Figure 7.** Daily mean $PM_{10}$ values (a) and dust load (b) calculated over the March-April 2020 period. Dashed lines represent values calculated from the counts of the optical particle counters in Bologna (BO), Trieste (TS), and Mt. Cimone (CMN), while solid lines represent concentration values recorded in Bologna at two air quality stations from the network (Giardini Margherita GM, Porta San Felice SF).

## 3.3 Ancillary analyses

### 3.3.1 AERONET (AErosol RObotic NETwork)

Figure 8 shows the Aerosol Optical Depth (AOD) data retrieved from AERONET ground-based remote sensing aerosol network at the site of Venice (Acqua Alta Oceanographic Tower - AAOT; 45°19'N, 12°30'E) around the period of the mineral dust outbreak event herein analyzed. Figure 8 shows very clearly the peculiar, steadily high aerosol optical depth value in the period 28 – 30 March and the simultaneously low value of the fine mode fraction, confirming the abrupt change in aerosol optical properties due to the dust intrusions in agreement with the OPC data previously described and similar to previous observations of dust intrusions over Italy (Boselli et al., 2021; Brattich et al., 2015a; Romano and Perrone, 2016). The decrease in AOD with wavelength observed on 28 March (Fig. 7a), i.e., right at the beginning of the dust incursion over northern Italy, is smaller than in the following days, which is likely caused by the larger fraction of coarse particles present in the atmospheric column and to the correspondingly low fraction of fine particles (Fig. 8b).

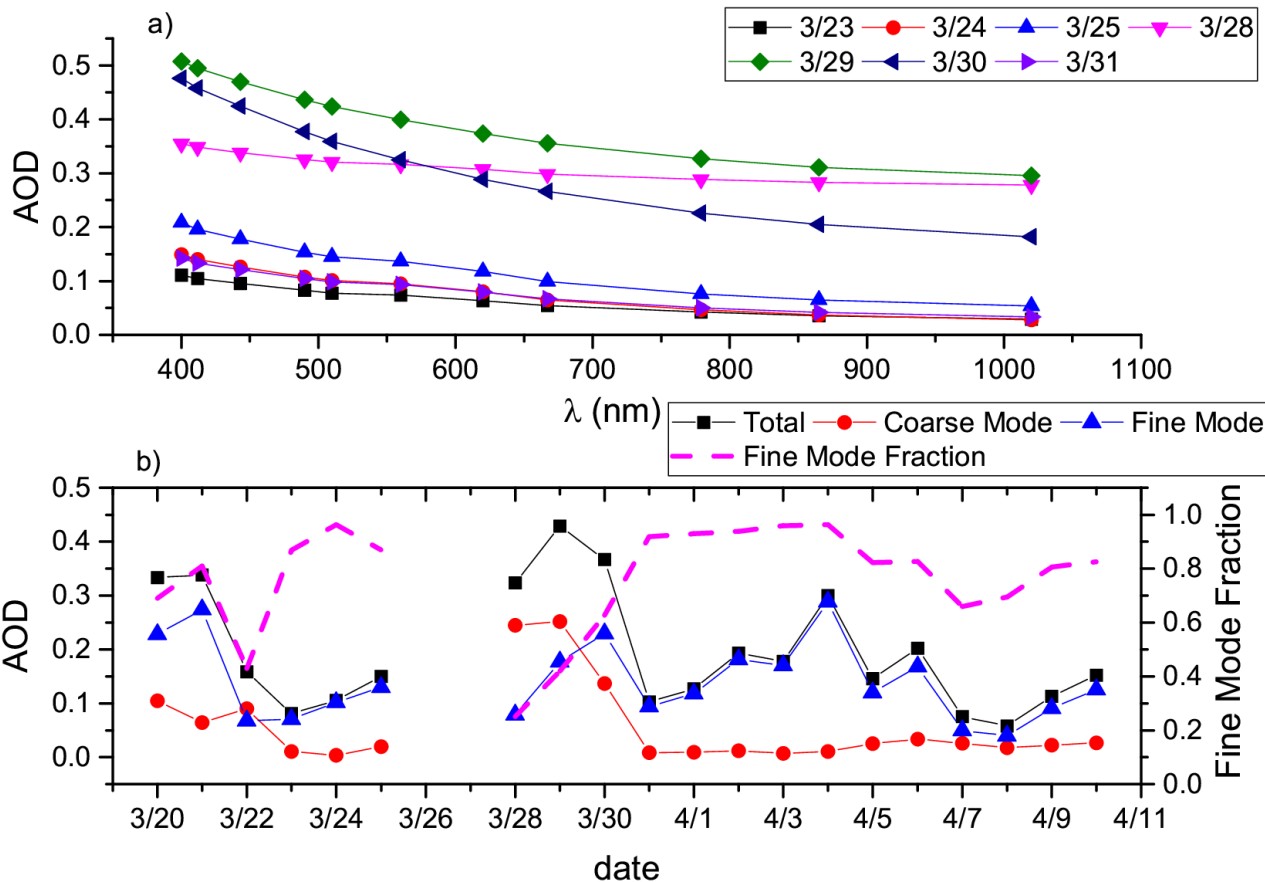

**Figure 8.** a) Aerosol Optical Depth (AOD) at different wavelengths and for different days of March 2020 detected at the AERONET site of Venice; b) time series of the total, coarse, and fine mode AOD, and fine mode fraction observed in March-April 2020 at the AERONET site of Venice.

Even the volumetric size distributions retrieved by AERONET indicate the prominence of the coarse mode, i.e., of particles with an aerodynamic diameter > 1 μm, on 28 and 29 March in agreement with this observation and with the particle size distributions from OPC readings in BO and TS (Fig. A5).

### 3.3.2 LIDAR Ceilometer

Images from a LIDAR ceilometer from ALICENET (Dionisi et al., 2018) located in Milan were employed to derive information on aerosol vertical distribution and of the dynamics of the atmospheric lowest layers. The sensor achieves excellent signal quality thanks to its stable wavelength and its Nd:YAG narrow-beam microchip laser operating in the 1064 nm range. In addition, its technology enables efficient daylight suppression and reduces temperature fluctuations to a minimum. Figure 9 shows the vertical profiles observed during the event, showing an approximate thickness of about 2 km above ground level. This observation indicates another peculiarity of this mineral dust event, which, differently from Saharan dust plumes typically reaching and shifting aloft between 1500 and 4000 m a.s.l. (Jorba et al., 2004; Soupiona et al., 2020), traveled at relatively low atmospheric altitude, reasonably due to the low temperatures at the source and to the reduced convective activity resulting from the higher latitude and season as compared to the Sahara desert. The relatively low travel height of the dust plume caused the gravitational settling of the heaviest particles in the proximity of the ground, in agreement with the size distribution data observed at Mt. Cimone as shown in Fig. A4c.

Figure 9 also shows that the yellow layer was somewhat deeper during the first three hours of 28 March than during the rest of the day; this observation likely indicates that early in the morning, the dust plume was capped by a cloud top, reasonably caused by the irruption of the easterly cold front towards the Po Valley. The layer was shallower from 4:00 UTC on. Signals of convective activity appeared between 10:00 - 15:00 UTC, as indicated by the more intense orange vertical swipes linked to the rising thermals. In the evening, the layer was even shallower due to the nocturnal cooling suppressing vertical motions. In general, the thinning of the mixing layer during dust transports, also confirmed by the analysis of the radiosoundings of the period, was linked to the high concentrations of aerosols in the troposphere, which reduced the amount of solar radiation reaching the ground, in turn reducing sensible heat fluxes that drive the diurnal evolution of temperature and the PBL (Li et al., 2017; Pandolfi et al., 2014; Salvador et al., 2019). From 20:00 UTC on, 2 km-high clouds appeared, probably linked to the arrival of the cold front from north-northeast, while the yellow layer rose again because of wind-induced turbulence. The following day, thermals appeared again between 8:00 and 15:00 UTC, while the signal of Arctic air masses can be observed from 20:00 UTC on.

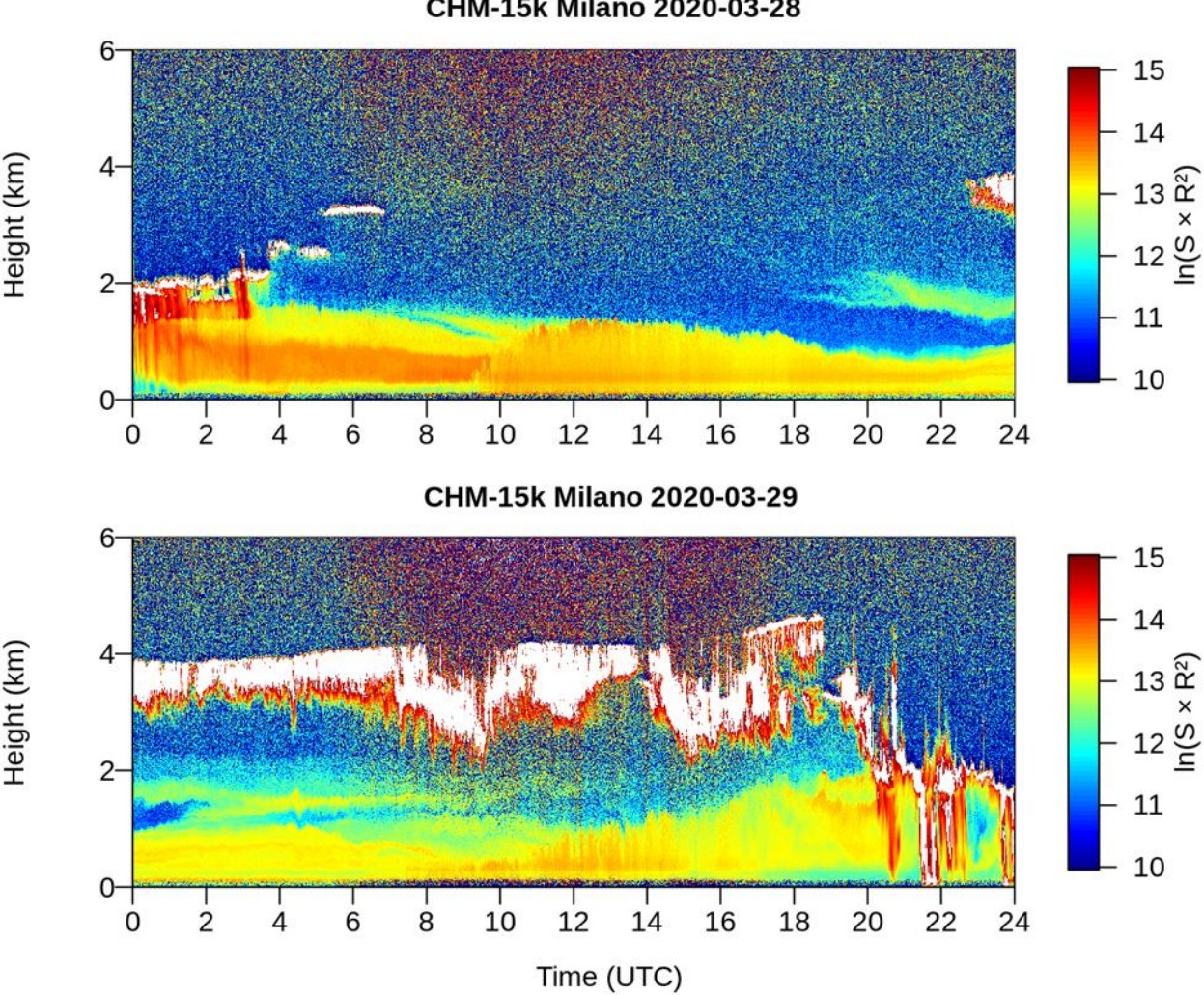

**Figure 9**. Vertical profiling from the automated LIDAR ceilometer located in Milan on 28 and 29 March 2020.

### 3.3.3 Chemical composition

The change in optical properties caused by the arrival of the dust plume is in agreement with the concurrent change in aerosol composition at the Bologna Gobetti station (Fig. 10). Indeed, the arrival of the dust plume from the Aralkum desert was connected with an increase in some major particulate components (i.e., nitrate, sulfate, elemental carbon) and of some trace elements (K, Pb, Na, Ca, Mg, Fe). This observation points out how not only crustal materials (Ca, Mg, and Fe) increased during the dust incursion dust, but also marine particles (Na) most likely from the Adriatic Sea and anthropogenic particles were entrained during the dust transit over the Balkans (Evangeliou et al., 2021). The entrainment of anthropogenic pollution and marine aerosols together with the crustal material originated from the Aralkum might have been favored by the low boundary layer previously observed, an aspect which has been put in connection with the exacerbation of the toxicological properties of dust aerosols (Pandolfi et al., 2014). However, it cannot be excluded that the presence of salt particles and heavy metals in connection with the dust incursion is due to their presence in

the dust source region, which contrarily to Saharan dust, is not a "true desert" but a dried sea where hazardous substances consistently accumulated for many decades (Groll et al., 2013). Conversely, the increase in crustal materials clearly indicates a typical composition of the mineral dust which in the source region is mostly characterized by quartz, calcite, and dolomite (Groll et al., 2019), while typical dust aerosols from the Sahara are mostly made by illite, kaolinite, and quartz (Formenti et al., 2014).

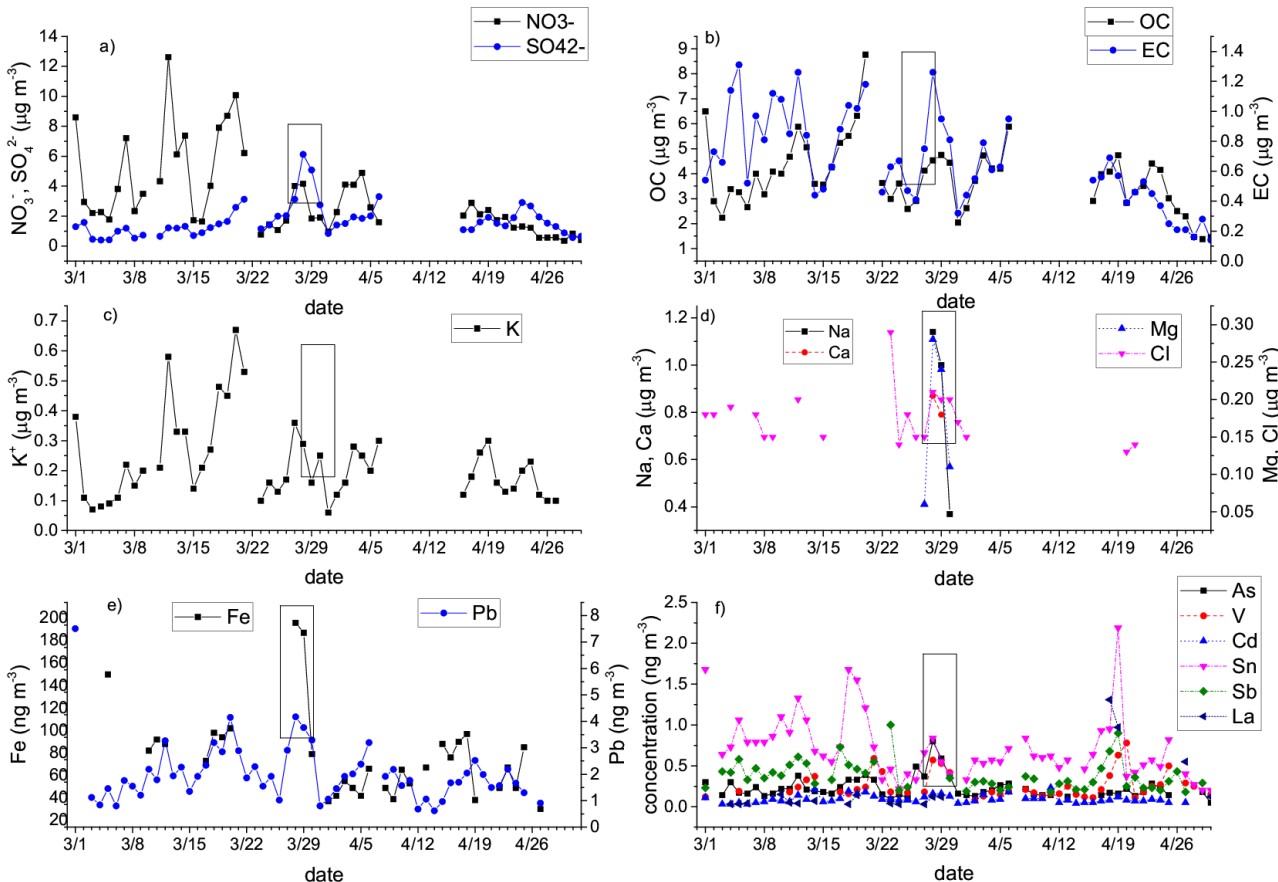

**Figure 10.** Concentration of major components and trace elements analyzed in PM$_{2.5}$ particulate samples at the Bologna Gobetti air quality station during March-April 2020 period: a) nitrate NO$_3^-$ and sulfate SO$_4^{2-}$; b) carbonaceous components organic and elemental carbon (OC and EC); c) potassium ion (K$^+$); d) Na, Ca, Mg and Cl; e) Fe and Pb; f) As, V, Cd, Sn, Sb, and La. The boxes enclose the dates of the dust incursion over Northern Italy.

## 4 Conclusions

This work presents a detailed investigation of a mineral dust event that reached northern Italy at the end of March 2020. The peculiarity of this event is mainly associated with its origin, which can be traced back to the desert region surrounding the Caspian Sea in central Asia. Its origin was confirmed by the concurrent analysis of the synoptic configuration together with satellite data of AOD and the calculation of back-trajectories for the study period. In particular, while it is well known how Southern Europe is often subjected to mineral dust incursions from the Sahara desert, transport of mineral dust originated in the Caspian area impacts frequently eastern Asia rather than western Europe.

The storm originated from the Aralkum desert, a favorable area for such conditions since the shrinking of the Lake in the last half-century. However, the peculiar meteorological situation of the second half of March, characterized by a presence of anomalous high Z850 geopotential in a region spacing from Siberia to the Atlantic ocean, contributed to creating a narrow corridor of flow between the Aralkum desert and northern Italy at lower levels (first 1.5-2 km of the atmosphere) for some consecutive days, along which and during which the dust was transported.

Data collected from three optical particle counters (OPCs), located at two different cities, Bologna and Trieste, and at one high-altitude, i.e., the WMO-GAW Mt. Cimone station, all in northern Italy, were used to study the event at high time resolution. The use of OPCs allowed characterizing the size distribution of this anomalous dust event accurately and understanding its features over time.

The results demonstrated that the transport of mineral dust mainly affected the coarse fraction concentration at all three sites, particularly those with a mean diameter higher than 0.7 μm. The fine fraction appeared negligibly affected by the central Asian mineral dust outburst, its concentration being similar to that observed before and after the event. Anyhow, the coarse fraction was affected by the event to the point that $PM_{10}$ concentration in Bologna exceeded up to twice - three times the WHO limit, in a period in which anthropogenic emissions were strongly reduced by the lockdown imposed in Italy to contrast the spread of the SARS-COV2 virus.

Overall, these results demonstrate how the concurrent analysis of multiple meteorological information and atmospheric composition data is needed to deepen our understanding of the various physicochemical processes connected with aerosols and their interactions. These results can be extremely important for their potential climatological implications owing to the potential increases in the frequency of dust storms resulting from the combined effects of increasing temperatures, increasing drought, and soil erosion. In particular, the fact that the prevailing mid-latitude westerly winds are weakening and migrating polewards as a result of anthropogenic warming, as suggested by recent works (Abell et al., 2021), might lead to changes in dust production and transport regions and to a potential increase of such kind of dust transport events from eastern sources.

**Data availability**

OPC observations and HYSPLIT back-trajectories used for this investigation are available upon request to the corresponding author (laura.tositti@unibo.it).

Synoptic maps are available on the wetterzentrale web archive: https://www.wetterzentrale.de/.

CAMS data are freely available on the Copernicus website at: https://ads.atmosphere.copernicus.eu/ or https://atmosphere.copernicus.eu/.

NASA AERONET observations are available at: https://aeronet.gsfc.nasa.gov/.

The lidar ceilometer images from the ALICENET are available by contacting the ALICENET project office at: alicenet@isac.cnr.it

Aerosol chemical speciation data are available on the open data portal at: https://dati.arpae.it/.

**Author contributions**

LT and EB designed the study. CC led the synoptic analysis. PM led calculation and analysis of HYSPLIT back-trajectories. AZ, EB, and AB developed the analysis methodology and led the analysis of observational data with contributions from all coauthors. AM provided and analyzed data for the Mt. Cimone station. SDS and FP contributed to the discussion of the results. LT, EB, and AZ wrote the manuscript with contributions from all coauthors.

**Competing interests**

The authors declare that they have no conflict of interest.

**Acknowledgments**

We gratefully acknowledge Dr. Marco Bellini and Dr. Fulvio Stel from ARPA FVG Regional Environmental Protection Agency for providing data at the Trieste via Pitacco and via del Ponticello stations. Open data of PM chemical speciation at the Bologna via Gobetti sampling site were provided by Emilia-Romagna Environmental Protection Agency (ARPAE). The HYSPLIT transport and dispersion model and/or READY website (https://www.ready.noaa.gov) used in this publication were provided by the NOAA Air Resources Laboratory (ARL). We wish to thank the NASA-GSFC AERONET team, and the PI Giuseppe Zibordi and his staff for establishing and maintaining the AAOT Venice site used in this investigation. We also thank the ALICENET (www.alice-net.eu) consortium for operating and maintaining the ALC network and for providing the Milan-ALC data used in this publication.

**Financial support**

This paper is published with the contribution of the Department of Excellence program financed by the Ministry of Education, University and Research (MIUR, L. 232 del 01/12/2016).

The study was also supported by a Research Grant from CARISBO Foundation, Bologna, Italy.

Monte Cimone station has received funding from the European Union's Horizon 2020 Research and Innovation Programme under grant agreement No 654109.

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

**Appendix A**

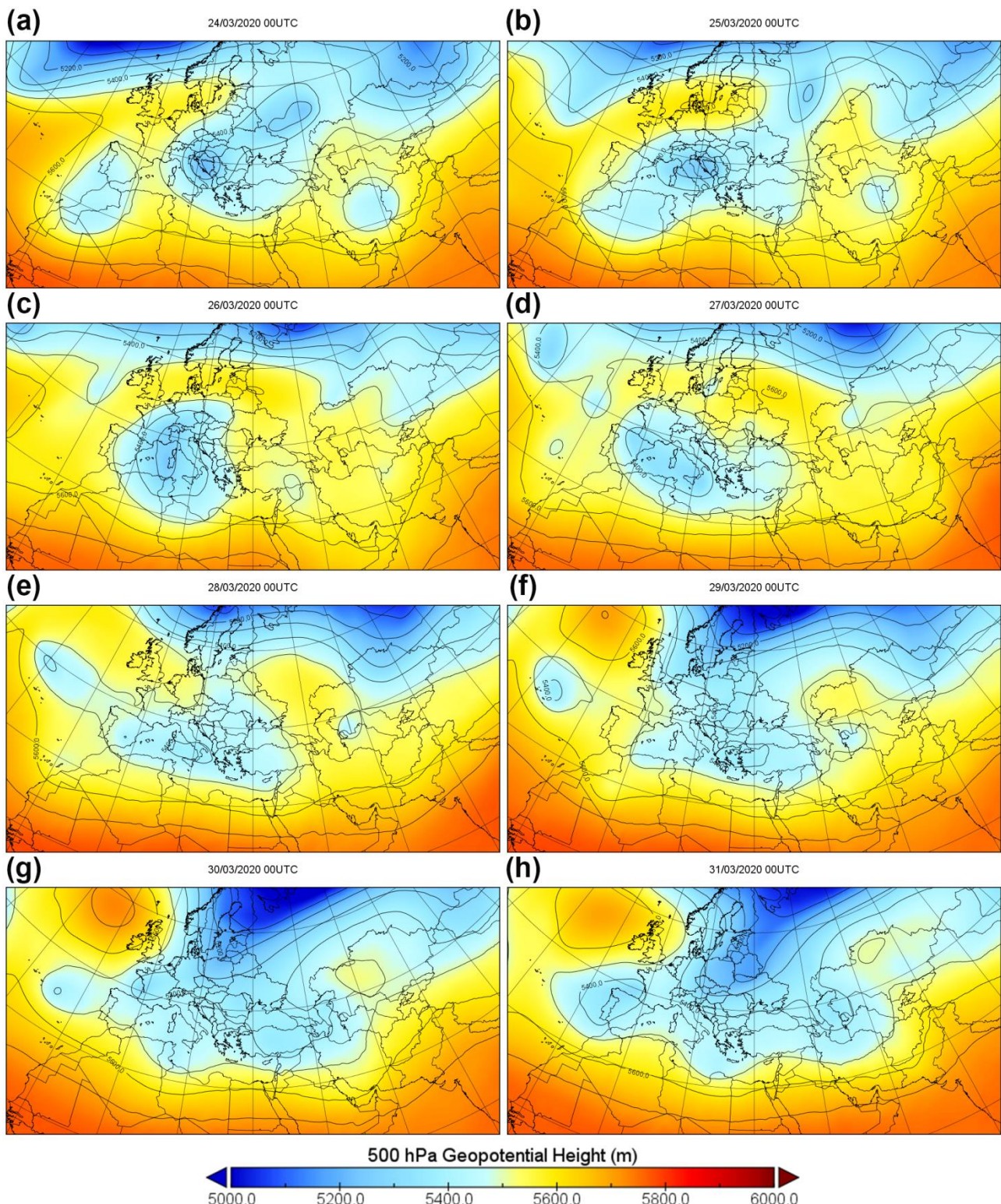

**Figure A1.** Geopotential at 500 hPa (in m$^2$ s$^{-2}$) maps (colors and isolines) relative to each day in the period 24-31 March at 00 UTC (a-h). The maps were created with Panoply starting from ERA5 reanalysis. Letters from a) to h) are referred to the days from 24 to 31 March 2020.

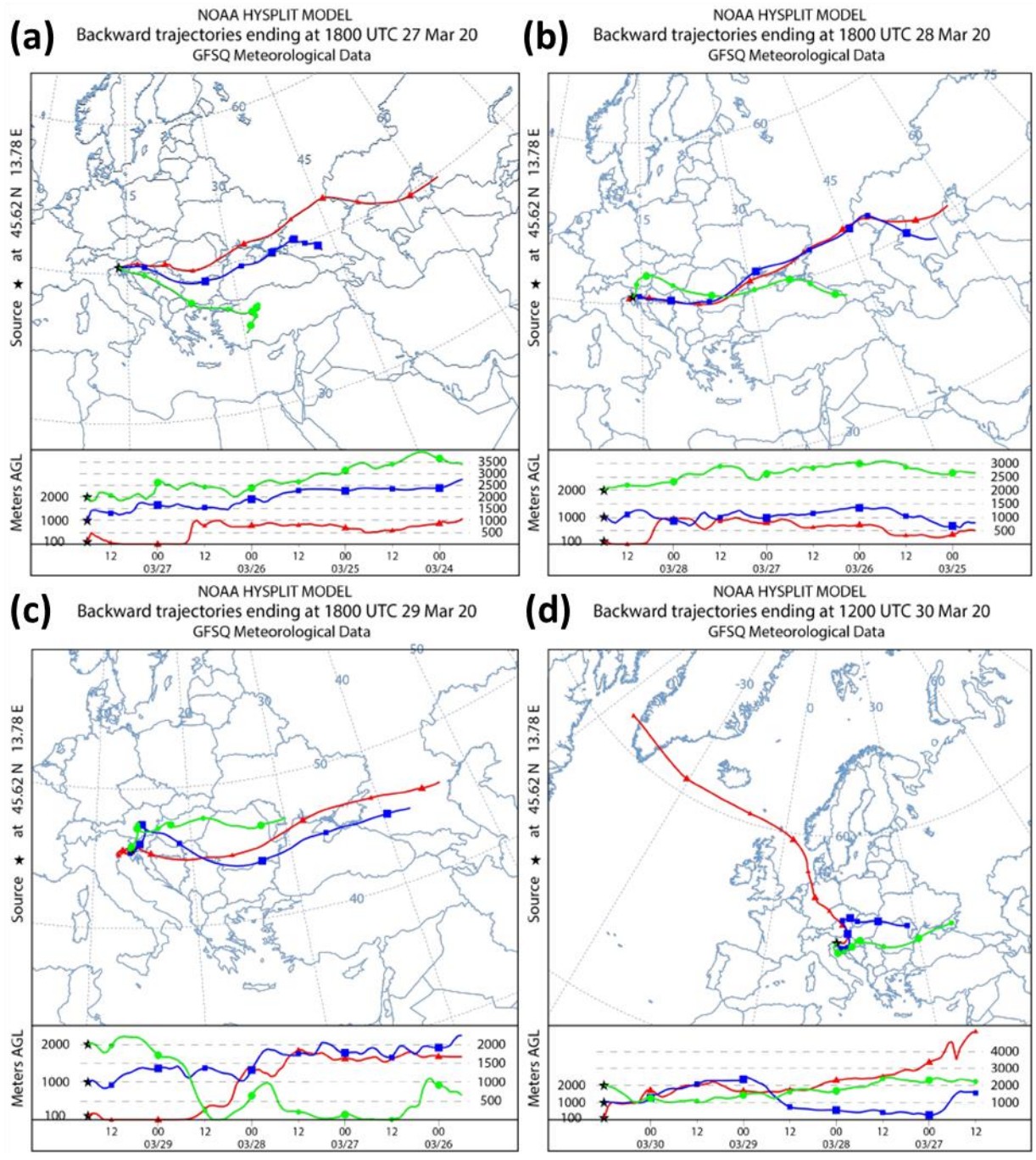

**Figure A2**. Back-trajectories (96-h backwards) ending at Trieste on a) 27 March 18:00 UTC; b) 28 March 12:00 UTC; c) 29 March 18:00 UTC; d) 30 March 12:00 UTC

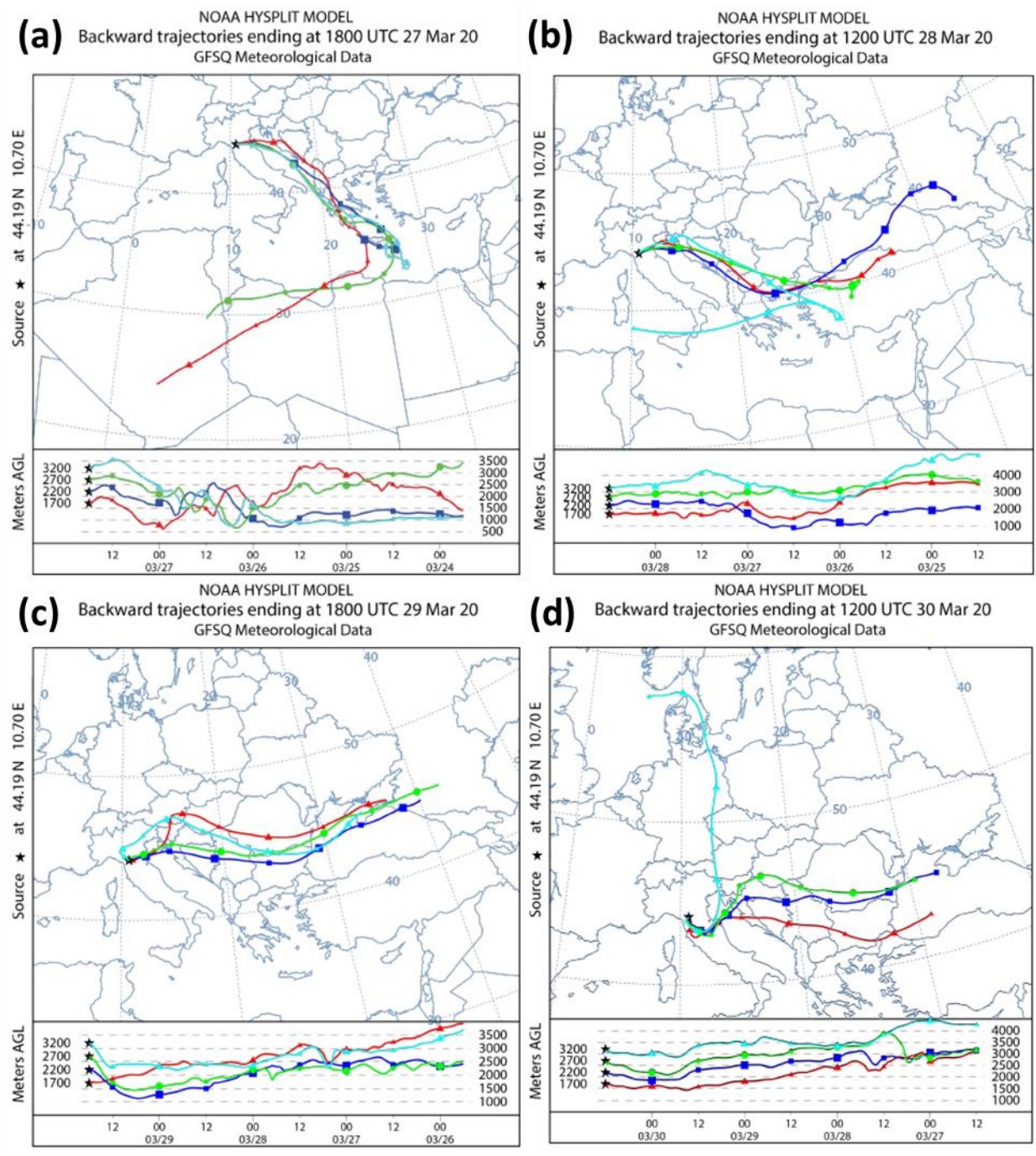

**Figure A3**. Back-trajectories (96-h backwards) ending at Mt. Cimone on a) 27 March 18:00 UTC; b) 28 March 12:00 UTC; c) 29 March 18:00 UTC; d) 30 March 12:00 UTC

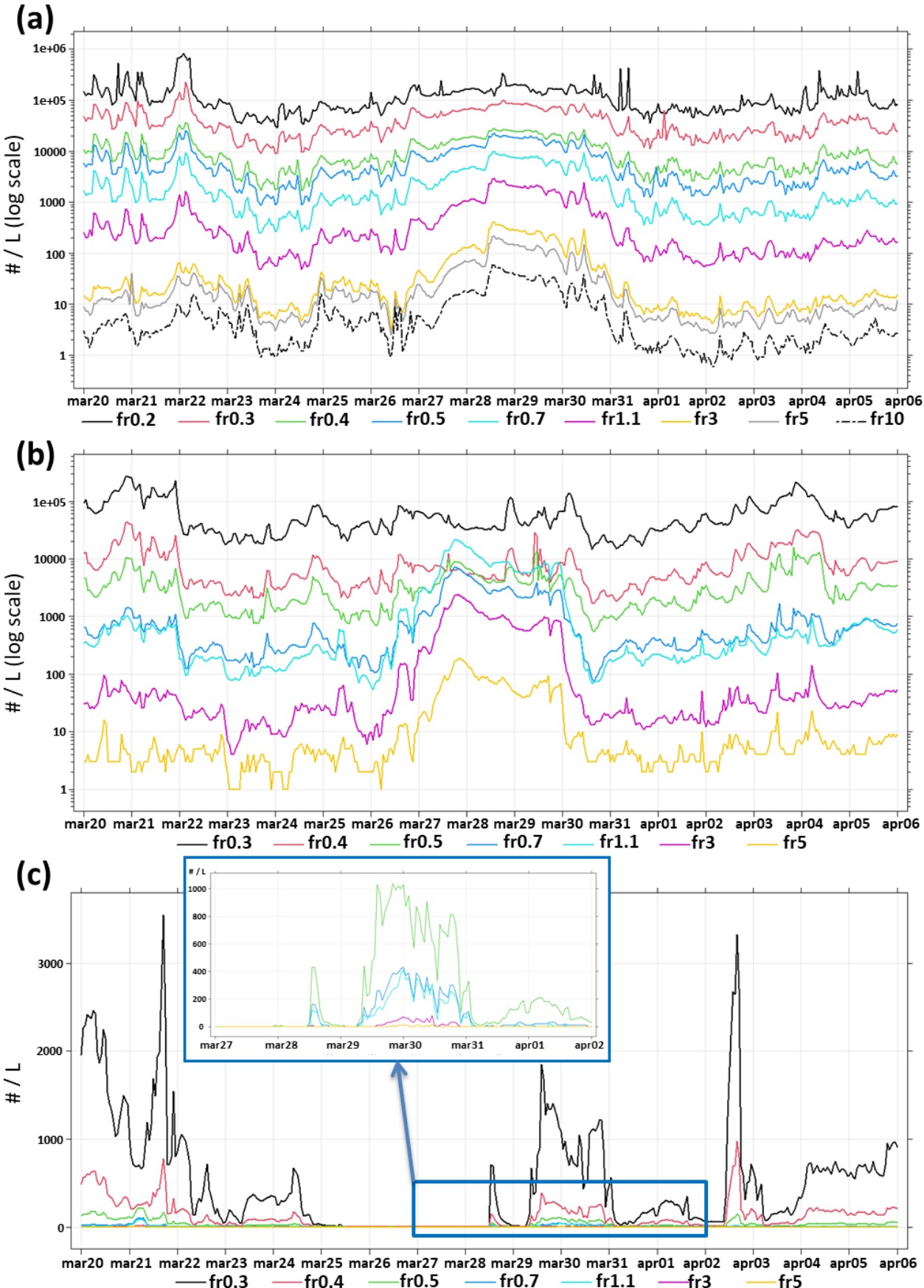

**Figure A4.** Temporal trends of all fractions for the three OPCs: a) LOAC (Bologna); b) FAI (Trieste); c) Mt. Cimone (the blue rectangle shows a zoom on coarser fractions). For BO and TS, values are reported on a logarithmic scale. Values are in Counts dm$^{-3}$ (# L$^{-1}$)

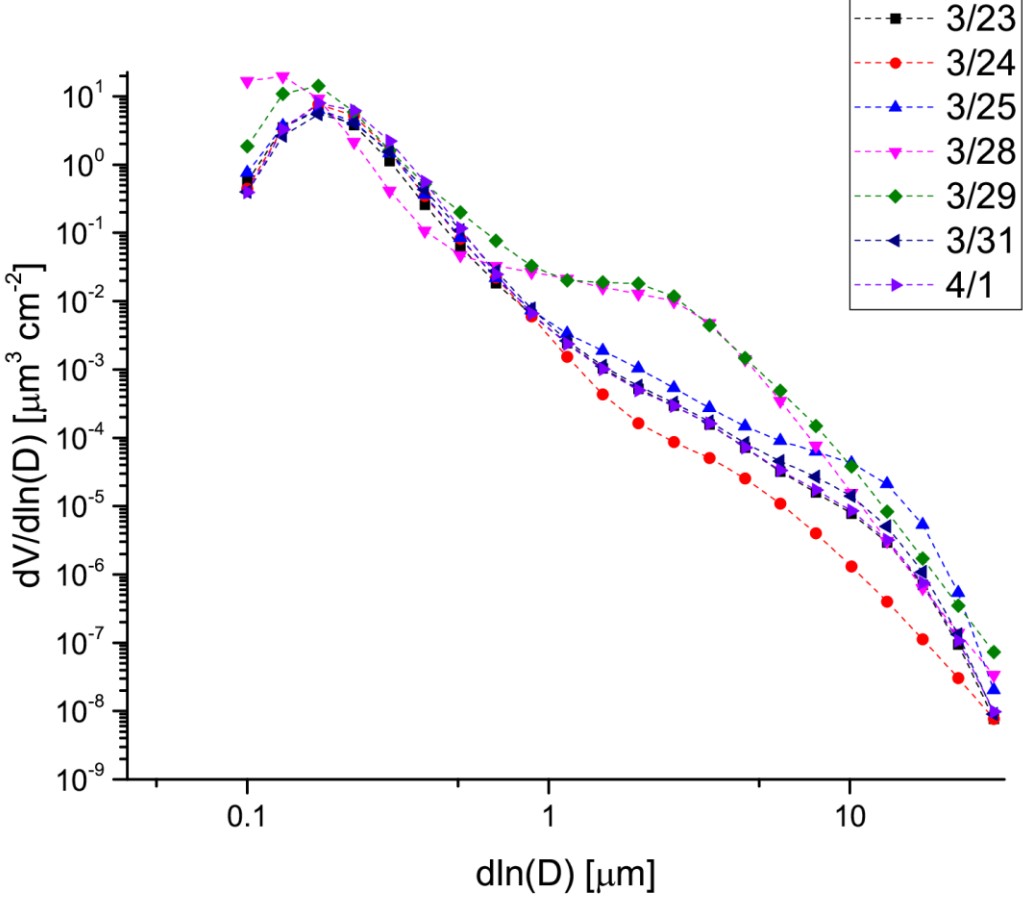

**Figure A5.** Volumetric size distributions retrieved by the inversion algorithm on days prior (23, 24 and 25 March), during (28 and 29 March), and after (31 March and 1 April) the arrival of the Caspian dust at the Venice AERONET site.