# Peer review of "Development and evolution of an anomalous Asian dust event across Europe in March 2020"

_Atmospheric Chemistry and Physics, 2021_

## Author Comment (AC1)

**Reply to the RC1's comments on "Development and evolution of an anomalous Asian dust event across Europe in March 2020" by Tositti L. et al.**

Manuscript Ref: acp-2021-429

We thank the reviewer for his/her useful comments. Below are the reviewers' comments in italic followed by our replies in blue text.

**General Comments**

*This study examines an interesting dust event that originated from the deserts in central Asia, most probably from the dried playas of the desiccated Aral Sea (Aralkum desert) and after following an unusual westwards movement affected the northern part of Italy 3-4 days later. This is an interesting dust episode affecting Europe, mostly because of the trajectory of the dust plume, since dust storms originated from Central Asia usually affect the eastern part of Iran, Afghanistan and/or they are transporting eastwards affecting Pamirs, Tian-Shan mountains and western China (Li et al., 2019, AtmosRes; https://doi.org/10.1016/j.atmosres.2019.06.013), Zhang et al. (2020; https://doi.org/10.1016/j.atmosenv.2020.117734) and references therein. Therefore, authors should give more emphasis on the uniqueness of this event.*

We agree with the reviewer and we have added more emphasis about the exceptionality of the event in the Introduction section at lines. The relevance of the findings and its potential implications in a climate change scenario was also further highlighted in the Conclusions section at lines.

*However, the dust source area for this event is not shown in the current figures and authors should provide at least satellite imagery for this dust event over Central Asia, at least for its genesis and transport. Another important issue is the comprehensive examination of the meteorological patterns that triggered and transported towards the west this dust event. The synoptic meteorology maps should definitely cover the central Asia and authors should explore the role of the Caspian Sea High (and the Siberian anticyclone) on this dust event.*

We modified the synoptic and the $PM_{10}$ maps (now Figures 1 and 2 in the main text, and A1 in the Appendix) to include the Central Asia region. We did not add satellite images during the days of the dust event, in particular over Italy, since some clouds obscured or partially filtered the satellite view. We used instead Panoply maps obtained by ERA5 reanalysis.

*Many previous works have shown that the Caspian Sea high plays an important role in dust activity and transport of dust plumes over central and southwest Asia (i.e. Kaskaoutis et al., 2016, Glob. Plan.Change; https://doi.org/10.1016/j.gloplacha.2015.12.011), Labban et al. (2021; https://doi.org/10.1002/joc.6611), Hamidianpour et al. (2021;*

*https://doi.org/10.1016/j.atmosres.2021.105711) and references therein. Furthermore, the Caspian Sea High in combination with the low over the North Sea, create the North Sea –Caspian Pattern (NCP) that significantly modulates climatic conditions over the Middle East and the Mediterranean, which may also affect the dust activity (you may see Brunetti and Kutiel, 2011; Nat. Hazards), Kutiel and Benaroch, (2002; Theor. Appl. Climat.).*

We thank the reviewer for his/her suggestion. We have included the detailed explanations above in the Introduction section.

*So, more emphasis should be given in satellite monitoring of the dust plume and transport and on the meteorological dynamics that carried the dust plume toward Balkans and Italy. On the other hand, the structure of the paper should be increased at several parts and some texts need revision. In the analytic comments below, I emphasize with specific comments to parts/lines of the paper that need revision and/or more discussion and clarification.*

We thank the reviewer for his/her suggestions. In the following we provide a point-by-point response to the suggestions provided.

**Specific comments**

*Lines 33-34: This cannot be a likely consequence from this dust storm, since the PM and pollution levels are not high. You may insist more on modulation of dust aerosol properties as a consequence of the mixing processes and not to health effects.*

As we have shown in Section 3.2.2, the PM levels increased consistently as a consequence of the dust storm, reaching values higher than 100 μg m$^{-3}$ recorded both in Trieste and in Bologna sites. As documented later in Section 3.3.2, the high concentration of aerosols in the troposphere decreased substantially the amount of solar radiation reaching ground, which in turn modified the diurnal evolution of the PBL in agreement with previous studies on Saharan dust transport. Therefore no changes are made in this respect.

*Lines 51-52: Here, you totally ignored the deserts in central Asia, China and Mongolia.*

We thank the reviewer for pointing this out. We have added other deserts in central Asia in the Introduction section.

*Line 65: Does it mean that these meridian pressure gradients are due to African Highs and south European lows? I do not think that this is right. For example, Sharav cyclones in north Africa are low-pressure systems moving across north Sahara. This sentence should be revised and better clarified if authors believe that discussing these dust mechanisms is required.*

We thank the reviewer for pointing this out to us. We have revised the sentence to better clarify the mechanisms that promote Saharan dust transports to the central Mediterranean in different seasons in the Introduction section.

*Lines 67-68: This is mostly right, but apart from the regional scale mechanisms large-scale dynamics, like NAO variability, also affects the dust activity across the Mediterranean. You may see a recent paper by Kaskaoutis et al. (2019, Atmos.Res.) and several references therein about these phenomena, as well as about Sharav cyclones (e.g. Bou Karam et al., 2010, J.G.R.).*

The reviewer is right, and we have added a reference to the NAO control on dust activity across the Mediterranean including the paper from Kaskaoutis et al. (2019).

*Line 84-86: For this statement, you may see the recent review study by Li et al. (2021; Geoscience Frontiers, 101180, https://doi.org/10.1016/j.gsf.2021.101180).*

We thank the reviewer for pointing out this review to us; the reference was added as suggested.

*Section 2.1. The measured particle size range from the various OPC instruments should be mentioned in Section 2.1.*

Measured particle size range of the three OPCs used in this work were added in Table 1 (Section 2.1).

*Line 131: According to satellite imagery (worldview website), the dust event was initially started over Aralkum on 24 March 2020. So, the beginning day cannot be considered as that when the dust plume affected Italy i.e. 27 March.*

We agree with the reviewer and we have changed it to "the beginning of the dust transport event to northern Italy".

*I would recommend to move the section 2.2 after the measurements description at the end of section 2.*

The old Section 2.2 (Synoptic scale conditions) was moved after the OPC Section. Now the OPC section is the 2.2 and the synoptic one is the 2.3. We decided not to move it at the end of section 2 because we wanted to maintain the same order in the "Materials and methods" and in the "Results and discussion" sections. However, as it was asked also by another referee, the "Sampling sites" and "OPC" sections are connected and the "Synoptic" section does not fit between them.

*The wind regime is the most important parameter in order to analyze the synoptic meteorology prevailed during the unusual dust-plume pathway. Therefore, vector winds should be superimposed in the already existed meteorological figures or should be plotted separately.*

We modified Figure 1 (now Figure 2) adding the wind vectors.

*Section 2.4. What's the instrument for PM10 measurements?*

We measured $PM_{10}$ starting from the OPC data. We added details about the algorithm used for such computation in Sections 2.2 and 3.2.2.

*Before the synoptic analysis (Section 3.1), authors should provide some satellite imagery for the dust emissions over Aralkum and the transport of a dust plume towards the Black Sea and the Balkans on the next days. By analyzing satellite imagery, I saw that the dust plume originated from Aralkum (mostly) on 24 March. These figures are absolutely necessary for somebody to follow the whole material. Unfortunately, there was extensive cloudiness over central-north Italy and western Balkans that obscured any dust plume towards Italy. This can be analysed via CAMS reanalysis (as authors did in Fig. 1).*

We added CAMS reanalysis together with horizontal surface winds in Figure 2 for the whole period of dust emission and transport to northern Italy (24-31 March 2020).

*Line 243: Why likely? Are you not sure about the source region of the examined dust event? Furthermore, the area that you described here is a vast arid-desert area with several active sources of dust like Karakum, Aralkum, Kyzylkum, Caspian depression, etc. You may see Shi et al. (2019, Atmos.Environ; doi: 10.1016/j.atmosenv.2019.117176) for more details.*

We have removed the term "likely" since our analysis provides no doubts regarding the source region of the dust event.

*Line 248: "with winter and spring as the driest seasons". I do not think that this statement is right. previous works over the region show that winter and spring are the wettest seasons. You may see Li et al. (2019, Atmos.Res.; https://doi.org/10.1016/j.atmosres.2019.06.013) and Li et al. (2021; https://doi.org/10.1016/j.gsf.2021.101180).*

We thank the reviewer for pointing out this typo. We have corrected adding a reference to the papers cited by the reviewer.

*Lines 248-249: You do not provide specific reasons for the analysis of the meteorological patterns here. The only reason is that you just selected the dusty period, and this is absolutely logic.*

We agree and we have modified the sentence as follows: "In order to analyze the weather patterns during the event, we have decided to analyze the pressure and wind fields at the sea level (SLP) and the geopotential height at 500 (Z500) and 850 (Z850) hPa in the period from 20 March to 30 March 2020.

*Lines 250-253: The synoptic maps should be redrawn, since they do not cover the dust source region, which is central Asia. There is coverage of the north Atlantic, but not of the source region. So, the reader cannot see the specific meteorological conditions that dominated over the area. Second, the PM10 maps correspond to CAMS reanalysis and are not satellite observations as mentioned here.*

We deepened the discussion and we added several other maps. Specifically, the revised figures are now the following: new Figure 1 contains the Z850 maps, Figure 2 the PM10 and the surface horizontal wind. Moreover, and in the supplementary material there is a new figure (Figure A1) containing the Z500 maps. Regarding the dates, the most important period in which the dust transport is active is between 24-31 March, so we have produced and described the maps during this time period. For the previous days, we provided only a summary description in the first part of the analysis.

*In general, The synoptic analysis should be rewritten in view of the new synoptic maps and emphasis should be given also over central Asia. One of the most important meteorological factors that determines the climate, dust activity and atmospheric circulation over the region is the Siberian High and its expansion over the Caspian Sea in late spring and summer. You may see previous recommended works by Li et al. (2019, 2021; Labban et al., 2021, Intern.J. Climatology; Hamidianpour et al., 2021, Atmos.Res.). So, the position and intensity of the Siberian/Caspian Sea High is very important to be discussed here. Authors may also see the anomaly from the mean climatology of the MSLP.*

In the text, we have indicated more clearly, during the days 20-26 March, the conditions over Caspian deserts, in which the dust originated. We also added the recommended references.

*For the AERONET analysis some few more discussions and references to previous works about modification of columnar aerosol properties during dust events in Italy should be added.*

References to previous works about modification of columnar aerosol properties during dust events in Italy was added as suggested.

*I believe that the last section 3.3.3 should be omitted and more analysis should be given in the previous ones. There are no obvious differences in the chemical composition of aerosols due to arrival or the dust plume. Especially, carbonaceous aerosols, sulfate, nitrate and other trace elements that are mostly related to anthropogenic pollution do not exhibit notable peaks that can be directly attributed to the dust plume. I would recommend to skip this section.*

In this section we have documented that the dust incursion was connected not only with an increase in crustal elements such as calcium and magnesium, but also of other elements and compounds likely originated from the mixing of the dust with atmospheric pollution and marine aerosols, as

facilitated by the low mixing layer height which we documented in the previous section. Likewise, we think that in spite of not being the core of this paper, this analysis represents a nice complement of the rest of the paper. No change is made.

*Line 481: There was not a change in dust optical properties. The aerosol optical properties were changed due to arrival of the dust plume.*

We agree with the reviewer and we have changed the sentence to "The change in optical properties caused by the arrival of the dust plume…"

*Line 482: That's the first time that Aralkum is referred as the dust source region. This analysis should be performed before.*

We agree with the reviewer that this is the first time that the Aralkum desert is referred as the dust source region. However, the chemical analyses were not performed by our group and the data were obtained by ARPAE, the regional agency for environmental protection. We decided to focus our work on the data obtained and analyzed by our group, such as the OPC ones, while we used chemical (and also AERONET and LIDAR) data as "ancillary analyses", whose aim is to deepen the description of the event without focus the attention on these data.

*Line 482-487: This is a very long sentence which should be split into two or three.*

We agree with the reviewer. We have modified the sentence as follows: "Indeed, the arrival of the dust plume from the Aralkum desert was connected with an increase in some major particulate components (i.e., nitrate, sulfate, elemental carbon) and of some trace elements (K, Pb, Na, Ca, Mg, Fe). This observation points out how not only crustal materials (Ca, Mg, and Fe) increased during the dust incursion dust, but also marine particles (Na) most likely from the Adriatic Sea and anthropogenic particles were entrained during the dust transit over the Balkans (Evangeliou et al., 2021)."

*Conclusions may be reconstructed giving more emphasis on the source region and specific meteorological conditions that carried dust from central Asia to north Italy.*

We deepen the Conclusions, including a synthesis of the synoptic conditions, in which the source regions were explicitly mentioned.

**Minor comments**

*Lines 18-21: You may also include the dates in the abstract.*

*Line 23: "plume" instead of "cloud"*

*Line 50: Revise as "dust sources" or "dust-source regions"*

*Line 84: replace "reminded" with "mentioned".*

*Line 100: Revise this sentence "during transport" does not make good sense.*

*Line 111: Revise as "was analyzed or investigated or studied.."*

*Lines 113-114: This sentence makes no sense. Something is missing. It should be rephrased.*

*Line 134: Reconstructed is not the appropriate term here. The sentence should be revised.*

*Line 139: Delete double use of "we have"*

*Line 156: 2.3 Heading. Revise as "particle"*

*Lines 162-165: This sentence needs revision.*

*Line 177: You should revise this sentence. The equation 1was not previously indicated.*

*Line 253: I think this is 25 March.*

*Line 342: "evolution" or "variation" instead of trend. You did not examine the trends here.*

*Lines 348-353: This sentence is very long and it should be revised and split into two. You may use Italian references for COVID-19 lockdown in Italy.*

*Line 389. "settling" instead of "selection".*

*Lines 391-392: Define the time frame in order to make it clear.*

*Line 420: Add "and" between the concentrations.*

*Figure 6 caption. In this figure there are two measurement sites in Bologna, not three.*

*Line 467, 476: You should complete these sentences.*

We corrected all these typo and rephrasing revisions

---

## Author Comment (AC2)

**Reply to the RC2's comments on "Development and evolution of an anomalous Asian dust event across Europe in March 2020" by Tositti L. et al.**

Manuscript Ref: acp-2021-429

We thank the reviewer for his/her useful comments. Below are the reviewers' comments in italic followed by our replies in blue text.

**RC2**

*This study presents an interesting dust event that originated from Central Asia and was evident in parts of Southern Europe, with a specific focus in Italy. The event is discussed mainly by means of particle number densities and particle size distributions, derived from optical particle counters located at three different sites, namely Bologna, Trieste and Mt. Cimone. While the paper is overall well written, some parts of it need revisions and the syntax could be improved.*

We thank the reviewer for his/her overall appreciation of our work and for the useful comments provided improving its quality. In the following we reply specifically to each of the comments raised.

*Specific comments:*

- *Section 2.1: Since the instrumentation is not uniform for all three sampling sites, I would recommend summing up all the relevant information in a table.*

We added a Table (Table 1) in Section 2.1 with the relevant information for the three OPC

- *Section 2.2 would fit better in the end of Section 2.*

We moved old section 2.2 after the OPC-section (now it is section 2.3). Our original idea was to put the descriptions of the methods and of the results in the same order of the description of the results, but we agree with the reviewer that the "synoptic conditions" paragraph does not fit between the "sampling sites" and the "OPC" sections.

- *The authors should explain their choice of using the total, fine and coarse mode AOD at the 500 nm wavelength.*

We have inserted a reference to the work of O'Neill et al. (2003) explaining in detail the spectral deconvolution algorithm applied to retrieve total, fine and coarse mode AOD at the standard reference wavelength of 500 nm. Now the sentence in Section 2.5 reads: "Total mode, fine mode, and coarse mode AOD at 500 nm (standard reference wavelength) were computed using a best-fit second-order polynomial according to the spectral deconvolution algorithm (SDA) by O'Neill et al. (2003). "

- *Section 3.1 needs to be rewritten given more focus to the synoptic conditions above Central Asia. In addition, Fig. 1 needs to be revised and include the dust source region.*

We deeply modified Section 3.1 and Figure 1 with more emphasis on the synoptic conditions above central Asia. We also enlarged the number and type of maps reported the old Figure 1, and we included now a second Figure (Figure 2) to make the results clearer.

- *Both sections 3.3.2 and 3.3.3 would benefit from a more in depth analysis.*

We added a more in-depth description of the chemical data (Section 3.3.3). However, both LIDAR and chemical data were not recorded by us and are not the main focus of the paper, which are the synoptic and OPC analyses. These sections, indeed, are reported as "ancillary analyses" and are used only to give a more general description of the event. Therefore we decided not to further deepen this part of the discussion.

- *A.2(a): The backtrajectories ending at 1.7 km and 2.7 km altitude originate from the Sahara region. In line 330 the authors state that the air mass "it did not give origin to visible peaks in the OPC time-series". Please elaborate more on that statement. Perhaps a peak would is visible for other fractions.*

Figure A4 (A3 in the previous version) shows all the fractions registered by the OPCs, and in the case of Mt. Cimone no peak in all fractions was observed on March 27, the day for which those back-trajectories were calculated. We added a reference to Fig. A4 in that paragraph and a more deepen explanation has been added in Section 3.2.1: it is likely that no peak was observed in March 27 because a strong north-easterly wind flowed over Mt. Cimone, making impossible for the OPC to register a particle increase.

*Minor comments:*

- *Please make sure that all acronyms have been defined properly in the manuscript (e.g., EUSAAR in line 129).*

All acronyms are now defined.

- *Lines 345-346: A reference for the statement would be useful.*

A reference for the statement was inserted.

- *Figures 5 and 9 are difficult to read.*

The figures quality have been improved.

---

## Author Comment (AC3)

**Reply to the RC3's comments on "Development and evolution of an anomalous Asian dust event across Europe in March 2020" by Tositti L. et al.**

Manuscript Ref: acp-2021-429

We thank the reviewer for his/her useful comments. Below are the reviewers' comments in italic followed by our replies in blue text.

This paper provides a detailed investigation of a strong dust event over Italy during March 2020. The most interesting fact is that the origin of this dust event is located in central Asia, which is a rare source of dust for Europe. The synoptic conditions before and during the dust event are also discussed. The evolution of the event is monitored by in-situ measurements from optical particle counters (OPC) at three stations located in northern Italy. These measurements are complemented by back-trajectories analysis, air quality products from the Copernicus Atmosphere Monitoring Service (CAMS), AERONET AOD, and vertical profiles from a Lidar Ceilometer. Some information is also provided regarding the particles' chemical composition. The paper is generally well written, however there are some points that must be clarified or better explained.

We thank the reviewer for his/her overall judgement of our paper. Below we have answered to each specific point raised by the reviewer.

Line 25: "supported by AOD (Aerosol Optical Depth) maps" You are not showing AOD maps in the manuscript. However, I think that the paper would greatly benefit by including a map of satellite derived AOD (such as from MODIS) or reanalysis AOD data from MERRA-2.

We decided not to use AOD maps from MODIS satellite because the satellite signal was disturbed by the presence of clouds over Italy during the event, thus only in March 28 the satellite images detected an AOD signal from the dust transit. For this reason, we decided to use the CAMS reanalysis data instead of using satellite or reanalysis AOD data, as suggested by the reviewer. Accordingly, we have modified the text as suggested by the reviewer (Section 2.3).

Line 106 "aerosol vertical behavior" please rephrase to "vertical distribution"

We modified the term as suggested.

Sect. 2.4 Please provide more information about the two air quality stations of the ARPAE Regional Environmental Protection Agency in Bologna

We added more information in the revised version of the manuscript in Section 2.4.

AERONET: Please mention if you used "all-points" or "daily averages" data.

The information that we used "daily averages" data was added in the revised version of the manuscript.

Line 244: what do you mean by "ecologically active"?

We removed the sentence.

*Lines:* 251-252: "derived from satellite observations" You used data from CAMS product, which merges model and observation data. Please rephrase to "derived from CAMS".

We modified the term as suggested from the reviewer.

Lines 253-267: You discuss the synoptic conditions during 20-24 March without providing the respective maps. I suggest to include the Z500 maps for the aforementioned period in a supplement/appendix. Please include in the supplement/appendix the maps for Z850 during the episode days. Also, consider including in Figure 1 the Z500 and PM10 concentrations maps for the 29 March.

At the beginning, we decided to include only a few maps, in order to limit the number of figures. Since all reviewers raised this point, we deepened the discussion and we added several other maps. Specifically, the revised figures are now the following: new Figure 1 contains the Z850 maps and Figure 2 the PM10 and the surface horizontal wind. Moreover, in the Appendix there is a new figure (Figure A1) containing the Z500 maps. Regarding the dates, the most important period in which the dust transport is active is between 24-31 March, so we have produced and described the maps during this time period. For the previous days, we provided only a summary description in the first part of the analysis.

Line 270: "with a double curvature in correspondence of the Balkans" ? What do you mean by "double curvature"? Please rephrase.

On March 25th, the flow is directed towards SW on western Black Sea, then veers towards NW on Serbia and Bosnia, and the veers again towards SW on Adriatic sea and Italy, thus with an "S-shaped" path (double curvature). However, since this behavior is well visible on the maps (especially on those reporting surface winds – see the new Figure 2), we have eliminated the subsentence.

*Lines 350-351: "besides the seasonal evolution pattern naturally leading to a decrease going towards the warmer season at this latitude" Please provide a reference.*

We agree with the reviewer that this sentence was not clear in the original manuscript. We have modified it to "besides the typical seasonal pattern at this latitude characterized by a decrease in the warmer season" and we have added a citation to Perrino et al., 2014.

**Fig. 4: What about the polar plot and the wind speed & direction time-series for CMN?**

We added a description of CMN polar plot and wind time-series at lines 469-480, with separate discussions relative to the different locations of Trieste, Bologna and Cimone, in which different behaviors correlated with the meteorological situation were founded.

You mention (line 375) that "the dust event was preceded and followed by weak winds, lower than 2 m/s". However, the time series in Fig. 4 does not support this statement. For example, for Trieste, between 25-27 March we observe a persistent NE flow (ws>10m/s). During 28-29 March the wind speed decrease, while it increases again in 30 March. Over Bologna the wind speed is much lower compared to Trieste and does not exhibit clear patterns. Moreover in March 28 (day with high dust concentrations) the mean wind speed seems to be lower than before the event. Please elaborate more on the results presented in Figure 4, taking also into account the previous discussion of the synoptic conditions.

The results shown in Figure 4 are better and deeper discussed and presented in the revised version of the manuscript, in Section 3.2.1, with some hypotheses justifying the differences among the behaviors in different stations.

Line 406: please provide a short description of the algorithm and its possible shortcomings.

We added some details and a reference about the algorithm in Section 3.2.2.

Line 408: You state that there is "a negative bias" with respect to the experimental values recorded by the Regional Environmental Protection Agency in Bologna. However, according to Fig 6a the solid lines representing the concentration values recorded in Bologna ARPAE stations are slightly above the blue-dashed lines (OPC data in Bologna). Also, the red dashed line (Trieste) seems to fits better with the ARPAE line. Is there a chance that something went wrong in Fig6? Are you sure that colors represent the appropriate time-series? Please check.

"Negative bias" means that the OPC-calculated values of  $PM_{10}$  in Bologna (blue dashed line) are underestimated with respect to the ARPAE stations (solid lines), as correctly shown in Fig. 6a (now Fig. 7a). Indeed, the blue dashed line is always lower than the solid lines. However, we have checked the data from the ARPAE stations and found a mistake in the previous figure. The revised version of the manuscript contains the corrected data for  $PM_{10}$  and dust load.

*Line 416-417: "while two relative maxima were detected on 27 March (36.4 \mug m-3) and 22 March (35.0 \mug m-3)" Please check the reported dates, they do not seem to be in agreement with the figure.*

These values are referred to the blue dashed line of Figure 6a (now Fig. 7a), which, besides the maximum due to the event, show a peak in March 22 and another relative maximum (less visible on the graph) on March 27. We pointed out in the text that those values are referred to OPC data.

Line 421-423: According to Fig 6, the Trieste time-series are shorter (20/3 - 6/3) than those of Bologna, (1/3 - 18/4). From witch dates the mean PM10 values (before and after the dust event) in Trieste are calculated? Averaging data from different dates can lead to non-comparable results between Trieste and Bologna regarding the pre- and post-event mean PM10 concentrations.

The daily means averaged for such comparisons are only those of the common period 20 March - 6 April: we added some specifications about it in the text. The longer time series reported in Fig. 6 (now Fig. 7) are used only to show the "baseline" of PM10 and highlight the peak due to the Aralkum dust transport.

*Fig.* 6, *caption:* "Bologna at three air quality stations from the ARPAE network" possible typo: three-> two

The typo was corrected in the revised version of the manuscript.

Sect. 3.3.1, AERONET: Please elaborate more on the spectral variation of AOD. For example, on 28 March the decrease of AOD with wavelength is smaller. This probably is caused by a larger coarse particles fraction.

Further information of the spectral variation of the AOD, together with references, were added to the section in the revised version of the manuscript.

*Line 484: "The change in dust optical properties". There is no change in dust optical properties. The aerosol optical properties changed due to the dust intrusion.*

We corrected the sentence in the revised version of the manuscript.

Sect. 3.3.3, general comment: Please elaborate more on the findings regarding the chemical composition, pointing also to the differences between the analyzed dust event and typical Saharan dust events.

We thank the reviewer for his/her comment. We have added more information on the findings about chemical composition in the revised version of the manuscript.

conclusions: please write 2-3 sentences mentioning the synoptic conditions resulting in the dust event.

We have included a synthesis of the synoptic conditions in the revised conclusions.

---

## Referee Report (RR1)

My previous remarks were properly addressed by the authors and I believe that the manuscript is improved. I have only a few minor comments.

l52: I think that the most correct spelling is "Kyzylkum" and not "Kyzilkum".

l121: "while multiple data based on remote sensing (satellites, LIDAR, and AERONET sun photometers)"
You didn't use satellite aerosol data, please rephrase to:
"while multiple data based on reanalysis (CAMS) and remote sensing (LIDAR, and AERONET sun photometers)"

l126-127 "Trend of AOD, aerosol vertical behavior distribution and chemical composition of the dust event from satellite platforms."
also, please rephrase removing the word "satellite".
L643: please rephrase removing the word "satellite". You employed CAMS reanalysis PM10 data and not satellite AOD.

Technical correction

Figure A4: If possible improve the quality of the embedded (small) figure.

---

## Author Response (AR2)

**Reply to the RC2's and RC3's comments on "Development and evolution of an anomalous Asian dust event across Europe in March 2020" by Tositti L. et al.**

Manuscript Ref: acp-2021-429

We thank the reviewers for their useful comments. Below are the reviewers' comments in italic followed by our replies in blue text.

**RC2**

*A few technical corrections:*
*• Lines 67-71: Consider switching the order of these two sentences.*

We switched the two sentences

*• Overall use past tenses when describing events that took place in the past.*

We corrected the text using past tenses where necessary

**RC3**

My previous remarks were properly addressed by the authors and I believe that the manuscript is improved. I have only a few minor comments.

*l52: I think that the most correct spelling is "Kyzylkum" and not "Kyzilkum".*

We corrected the typo

*l121: "while multiple data based on remote sensing (satellites, LIDAR, and AERONET sun photometers)" You didn't use satellite aerosol data, please rephrase to: "while multiple data based on reanalysis (CAMS) and remote sensing (LIDAR, and AERONET sun photometers)"*

We rephrased that sentence

*l126-127 "Trend of AOD, aerosol vertical behavior distribution and chemical composition of the dust event from satellite platforms." also, please rephrase removing the word "satellite".*

We corrected it

*L643: please rephrase removing the word "satellite". You employed CAMS reanalysis PM10 data and not satellite AOD.*

We corrected it

*Technical correction*
*Figure A4: If possible improve the quality of the embedded (small) figure.*

We improved the quality of the axes in the small figure of Fig. A4